# Structural conservation of HBV-like capsid proteins over hundreds of millions of years despite the shift from non-enveloped to enveloped life-style

**Sara Pfister[1], Julius Rabl[2], Thomas Wiegand[1,8,9], Simone Mattei [3], Alexander A. Malär[1], Lauriane Lecoq [4], Stefan Seitz[5,6], Ralf Bartenschlager [5,6], Anja Böckmann [4] ✉, Michael Nassal [7] ✉, Daniel Boehringer [2] ✉ & Beat H. Meier [1] ✉**

The discovery of nackednaviruses provided new insight into the evolutionary history of the hepatitis B virus (HBV): The common ancestor of HBV and nackednaviruses was non-enveloped and while HBV acquired an envelope during evolution, nackednaviruses remained non-enveloped. We report the capsid structure of the African cichlid nackednavirus (ACNDV), determined by cryo-EM at 3.7 Å resolution. This enables direct comparison with the known capsid structures of HBV and duck HBV, prototypic representatives of the mammalian and avian lineages of the enveloped *Hepadnaviridae*, respectively. The sequence identity with HBV is 24% and both the ACNDV capsid protein fold and the capsid architecture are very similar to those of the *Hepadnaviridae* and HBV in particular. Acquisition of the hepadnaviral envelope was thus not accompanied by a major change in capsid structure. Dynamic residues at the spike tip are tentatively assigned by solid-state NMR, while the C-terminal domain is invisible due to dynamics. Solid-state NMR characterization of the capsid structure reveals few conformational differences between the quasi-equivalent subunits of the ACNDV capsid and an overall higher capsid structural disorder compared to HBV. Despite these differences, the capsids of ACNDV and HBV are structurally highly similar despite the 400 million years since their separation.

Nackednaviruses comprise a recently discovered family of non-enveloped fish viruses[1] bearing similarities in genome organization and replication mechanism to the hepadnaviruses (hepatotropic DNA viruses), with human hepatitis B virus (HBV), a major human pathogen, as their most prominent representative[2]. Hepadnaviruses are small, enveloped DNA viruses, which replicate their tiny ~3 kb genomes by reverse transcription. Their replication strategy[3] involves co-packaging of one of the viral transcripts, the pregenomic (pg) RNA, with the multifunctional viral polymerase into newly forming nucleocapsids[4]. Therein, the pgRNA is reverse-transcribed into a partially double-stranded relaxed circular (rc) DNA, which is exported from the host cell in enveloped virions. Upon infection of a new cell, the envelope is stripped off, and the nucleocapsid delivers the rcDNA to the cell nucleus for conversion into a stable covalently closed circular (ccc)

---

A full list of affiliations appears at the end of the paper. ✉e-mail: a.bockmann@ibcp.cfr; michael.nassal@uniklinik-freiburg.de; boehringer@mol.biol.ethz.ch; beme@ethz.ch

DNA episomal minichromosome from which new viral transcripts, including pgRNA, are produced, completing the cycle.

Hepadnaviruses were long known to occur in a few mammals (genus *orthohepadnavirus*; type member HBV) and birds (genus *avihepadnavirus*; type member duck HBV [DHBV]), with hepatocytes as primary replication site. However, bioinformatic screens of genomic and transcriptomic databases[5] revealed the presence of HBV-like virus sequences in numerous other vertebrates, including amphibians, reptiles, and fishes (belonging to the genera *herpetohepadnavirus*, *metahepadnavirus*, and *parahepadnavirus*)[1,6,7], all sharing a common genome organization with *ortho-* and *avihepadnaviruses*. A typical hepadnaviral genome comprises open reading frames (ORFs) for the capsid, or core protein (Cp), the ~90 kDa polymerase (P) (occupying nearly three quarters of the genome) and, completely overlapping with the P ORF in a separate frame, for two or three envelope proteins; these encompass the transmembrane surface protein (S) plus N-terminally extended versions carrying one or two additional PreS domains. The envelope proteins of HBV and DHBV mediate infection of species-specific hepatocytes, and the same is expected for the other vertebrate hepadnaviruses.

Surprisingly, the bioinformatic analyses also identified a new virus family in diverse fish species whose members have similarly sized and organized genomes as hepadnaviruses but lack an envelope protein ORF separating them from the enveloped hepadnaviruses occurring in fish. The lack of an envelope led to their designation as nackedna (naked DNA) viruses[1]. Nackednaviruses do indeed employ a hepadnavirus-like protein-priming mechanism for reverse transcription[8], further supporting that both viruses had a common ancestor before they diverged about 400 million years ago, i.e. before the rise of tetrapods. It can thus be assumed that hepadnaviruses acquired their envelope protein ORFs at that time, likely by over-printing, i.e., by nucleotide substitutions in the pre-existing P ORF, which enabled expression of the envelope proteins from the same nucleotide sequence in a different frame[9].

Envelopes enable viral nucleocapsids to traverse host-cell membranes in either direction. For infection, the envelope proteins often interact with specific cell surface proteins; for HBV this is the hepatocyte-specific bile acid transporter sodium/taurocholate co-transporting polypeptide NTCP[10,11]. On the one hand, envelopes govern the infection mechanism as well as host and tissue tropism. Conversely, they allow for non-lytic progeny virion release. Hence, enveloped viruses must have evolved means to enable specific interactions between nucleocapsid and envelope during virion formation and the reversal of these interactions upon infection of a new cell.

Nackednaviruses are therefore likely to employ infection and virion release mechanisms that are, at least in part, different from hepadnaviruses, including different host and tissue tropism restrictions, as supported by the detection of nackednaviral RNA-derived sequences in various tissues from various fish species[1]. Typically, non-enveloped viruses, in essence naked nucleocapsids, alter their structure upon encountering cell surface receptors and/or environmental cues such as decreasing pH upon endocytosis. A well-studied example is poliovirus, where externalization of the N-terminus of viral protein 1 and the release of N-myristoylated VP4 from the virion occurs upon binding to the poliovirus receptor[12], enabling uncoating of the RNA genome for subsequent translation and replication. Release of naked progeny virions often occurs via cell lysis. Notably, the distinction between enveloped and non-enveloped viruses may be less strict, as indicated by the recent findings that non-enveloped viruses, including hepatitis A virus[13,14] and hepatitis E virus[15], also occur as membrane-cloaked quasi-enveloped particles; both can be infectious, and then use different intracellular trafficking pathways[16]. However, in contrast to enveloped viruses, quasi-envelopment does not require a virally encoded envelope protein. Regardless of this complication, it is highly conceivable that entry and secretion, as well as host range and tissue tropism restrictions are significantly different for nackednaviruses versus their enveloped hepadnavirus counterparts. On the other hand, in addition to enveloped virions also naked HBV capsids are released from infected cells via a different, Alix-dependent pathway[17]. The role of these naked HB nucleocapsids is not fully understood, however, it has been suggested that under certain conditions the naked nucleo-capsids may be involved in the transmission of the viral genome[17]. Hence, HBV might still have retained the ability to export naked capsids possibly associated with a conservation of the capsid structures even though enveloped HB virions are the main infectious species.

The capsid-forming core proteins (Cps) of classic hepadnaviruses come in two types; those of the ortho-hepadnaviruses comprise about 180 amino acids (aa). The first 140 residues constitute the N-terminal assembly domain (NTD), sufficient to form the capsid shell. Joined by a short linker (aa 140-149) follows a basic, arginine-rich C-terminal domain (CTD) that is crucial for nucleic-acid binding, regulated by dynamic changes in phosphorylation state[4,18]. Avihepadnavirus Cps encompass about 260 aa, with a similar but less strict separation into a ~180 aa NTD, including an ~40 aa extra extension domain, and a likewise highly basic CTD[19,20]. In both Cps, the NTDs adopt an all-α-helical fold in which two long central helices form an antiparallel hairpin; the hairpins from two Cp monomers associate into four-helix bundles, resulting in stable dimers as basic building blocks of the icosahedral capsids. The inter-dimer contacts are mediated by the most C-terminal helix (α5) and the immediately following sequence that folds back on α5 via a proline-rich turn; this structural module is often termed "hand region". Different from HBV, in DHBV, CTD residues also contribute to formation of the capsid shell[20]. For both viruses the overall architecture of most capsids conforms to $T = 4$ icosahedral symmetry, formed by 120 dimers each. This is made possible by the monomers adopting four similar quasi-equivalent conformations A, B, C, and D, giving rise to 60 AB dimers and 60 CD dimers, as confirmed by several X-ray and cryo-EM structures of CTD-less recombinant Cp variants and more recently by a 2.7 Å cryo-EM reconstruction of full-length Cp (pdb 6htx[21]). The Cp residues acting as hinges for adoption of the different conformations have been defined by solid-state NMR[22]. Especially for HBV, a minor class of capsids, both from recombinant Cp expression and in particles from HBV positive patient sera[23–25] and infected liver[26], display instead the 90 dimer $T = 3$ symmetry. The fraction of $T = 3$ particles increases when the C-terminal Cp end is truncated close to residue 140[27,28]. A 3.5 Å resolution cryo-EM structure for recombinant Cp149 $T = 3$ particles has recently been reported (pdb 6ui6[29]). The biological significance of the $T = 3$ HBV particles is unclear as they have not been isolated in sufficient purity for systematic comparison with their $T = 4$ counterparts during HBV replication. However, the much higher abundance of the $T = 4$ particles, also seen for DHBV[26], strongly argues that they are the relevant species in vivo.

Most nackednavirus capsid proteins comprise around 180 aa similar to orthohepadnavirus with a sequence identity of 24% for HBV vs. ACNDV). Secondary structure predictions suggested a similar α-helical fold as in the HBV Cp NTD, plus an additional 7 residue α-helix (termed α+) at the very N-terminus[1]. Bacterially produced Cp of the prototypic ACNDV, discovered in the species *Ophthalmotilapia ventralis* from Lake Tanganyika[30], in either full-length (174 aa) or in a C-terminally truncated form (aa 1-146) spontaneously assembled, like HBV Cp, into spiky particles, however virtually exclusively of $T = 3$ symmetry. Low resolution (8 Å) cryo-EM three-dimensional (3D) reconstructions confirmed the overall HBV Cp-like all-α-helical fold and suggested that the extra α + helix occludes the pores in the capsid shell[1]. However, the experimental resolution was too low to allow for a firm assignment of primary sequence with the discernable structural features.

In the current study, we employ a highly efficient bacterial ACNDV Cp expression system and solved the structure of ACNDV Cp by cryo-EM at 3.7 Å resolution. To facilitate the placement of the amino-acid

chain into the EM density we use positions of α-helices within the primary sequence, which we determine by solid-state NMR chemical shifts[31–33]. The resulting EM 3D structure of full-length ACNDV Cp enables clear tracing of the main chain and the visualization of individual large amino-acid sidechains. Our data confirm the exclusive formation of $T = 3$ particles, and they allow a direct residue-by-residue comparison with the HBV capsid. The NMR data demonstrate the structural homogeneity: all capsids have $T = 3$ and the same 3D structure (within detection limit of 5%). We investigated capsids under different buffer conditions in order to investigate pH-dependent structural changes including low pH as in the endosomal compartment, e.g., when the virus would enter a cell via endocytosis. No significant structural changes as a function of pH is detected neither by NMR nor by EM. The stability of the capsid, however, is significantly lower at lower pH and there is more structural disorder as demonstrated by NMR. We show that the ACNDV capsid is, despite the large evolutionary distance, structurally highly similar to the HBV capsid, however, with several notable differences.

## Results

The ACNDV capsid protein (Cp) was efficiently expressed in *E. coli* and the spontaneously formed capsid-like particles were purified by sedimentation velocity centrifugation through sucrose gradients analogously to recombinant HBV capsids[34]. Protein samples were analyzed by SDS-PAGE, transmission electron microscopy (TEM), and 2D carbon-carbon correlation solid-state NMR spectroscopy to demonstrate that samples were highly pure (Fig. S1).

Liquid-state NMR is not suitable for such large particles because the molecular tumbling is too slow, and we employed magic-angle spinning (MAS) solid-state NMR for an initial characterization of ACNDV Cp. All amino acids were uniformly $^{13}$C-$^{15}$N labeled. The capsids were sedimented into MAS rotors by ultracentrifugation alleviating the demand for crystals[35,36]. The 2D $^{13}$C–$^{13}$C correlation experiment with a DARR (dipolar-assisted rotational resonance)[37] mixing time of 20 ms for magnetization transfer in ACNDV capsids at pH 7.5 displays a 2D-fingerprint spectrum typical for an α-helical protein (Fig. S1c); key signals for the distinction between α-helical and β-sheet structures are the Alanine CA-CB crosspeak. All signal intensity is found around the α-helical shift values of 54.8/18.3 ppm and no intensity at 50.9/21.7 ppm (β-sheet). These two regions are known to be a clear indicator for secondary structure for alanine[38].

Resonances were sequentially assigned (Figs. S2a, S3a, for details see Methods) and secondary chemical shifts for identification of secondary structure elements (Fig. 1a) were calculated based on the NMR chemical shifts at pH 7.5 (experiments of Table S1, assignment, completeness Table S2). For assigned residues the secondary chemical shifts were calculated as: $(\delta(CA)-\delta(CA)_{random\ coil})-(\delta(CB)-\delta(CB)_{random\ coil})$ with $\delta(x)-\delta(x)_{random\ coil}$ being the difference between observed chemical shifts and random coil chemical shifts as tabulated in ref. [38]. If CA and CB shifts are both considered, α-helices and β-strands are reliably identified by secondary chemical shifts. Four or more positive values in a row suggest the presence of an α-helix, three or more negative in a row are indicative of a β-strand[39,40]. Information about helix positions and lengths, as indicated by colored background in Fig. 1a, was taken into consideration for building an initial protein model, which allowed for a fast de novo building of an initial structure. Furthermore, amide and HA hydrogens were assigned and provide a starting point for future experiments, e.g., for protein dynamics characterization[41]. The proton-detected hNH spectrum, which displays the correlation of hydrogen and nitrogen resonances, is shown in Fig. S3a.

The DARR spectrum of the pH 5.5 sample shows broadened peaks compared to pH 7.5 and additional weaker resonances in typical β-sheet chemical-shift regions of the spectra, notably new C′ resonances, upfield shifted when compared to pH 7.5, and cross peaks for leucine, valine and threonine, shifted downfield (Fig. S2c). These broad peaks likely represent denatured (sometimes also called aggregated) protein, and are also observed to appear after several months of storage at pH 7.5 (Fig. S4). By denatured we refer to protein that has partially lost or changed its secondary structure and not to a crowding together of capsids which we, in this paper, call aggregated. The latter hardly influences the NMR spectrum. The pH 5.5 sample is denaturing much faster (days to weeks) than the pH 7.5 samples, which themselves are more denaturation-prone than HBV Cp, and several other sedimented proteins[42]. Apart from signs for protein denaturation, three additional peaks were observed in the pH 5.5 spectrum, including resonances of histidine and isoleucine residues (Fig. S2d).

In order to further investigate the stability of the ACNDV capsids upon treatment with guanidine hydrochloride (GuHCl) we performed negative stain TEM. TEM images of the samples (Fig. S5) show that capsids in buffer containing 0.3 M GuHCl remain intact. However, with 0.5 M GuHCl capsids preparations become inhomogeneous with particles adopting a broader variety of different shapes. With 1.0 M no intact capsids can be identified anymore. For comparison, HBV Cp seems to be able to withstand higher concentrations of GuHCl than ACNDV Cp. It was for example found that GuHCl concentrations smaller than 1.25 M were insufficient for HBV capsid disassembly[43].

We used cryo-EM density maps of the pH 7.5 and 5.5 samples to build atomic-resolution structures of ACNDV capsids (Table 1, Table S8). The details of the reconstructions are shown in Figs. S6, S7, S8, S9 and the resulting structures in Fig. 2 and S9. The capsid is made up of 90 protein dimers to give $T = 3$ symmetry, with three molecules located within the asymmetric unit: chains A, B, and C (Fig. 2h, i). At the N-terminus, the ACNDV Cp sequence starts with a small α-helix (α+), which points towards the capsid exterior surface. However, the very first residues are not visible in the cryo-EM density map. The α2-helix is wrapped around the spike helices of the partner molecule in the dimer and the spikes are made up of a four-helix bundle within the dimer helices α3 and α4 in each monomer. Helix α5 together with downstream residues forms the hand region providing the inter-dimer contacts). The last 40 residues at the C-terminal domain (CTD) are disordered and no corresponding cryo-EM map density was discernible (only diffuse density).

For the pH 7.5 and the pH 5.5 structures, superimposed chains show that chains A and B are very similar while the N-terminus and the hand region is different for chain C compared to the other two chains (Fig. 3a and Fig. S10a). Comparison of the AB dimer, formed by the quasi-equivalent conformation of chains A and B, with the CC dimer, reveals that the four-helix bundles forming the spikes match well when superimposing AB with CC dimers, while the hand regions appear to be shifted (Fig. 3b and Fig. S10b). This rigid body like conformational shift between the quasi-equivalent capsid subunits observed in the cryo-EM structure agrees with the solid-state NMR measurements, which show few local conformational changes, as is further elucidated below. In contrast to HBV Cp, where 71 W and 88 Y sidechains interact with each other across the spike helices (Fig. S11f), only 60 H is part of a spike contact but no other large, hydrophobic sidechains (W, Y, F, or H) are involved in helix-helix contacts within the ACNDV capsid spike.

We observed that the asymmetric units at pH 7.5 and at pH 5.5 remain very similar—yet the capsid diameter is slightly smaller in our pH 5.5 model (about 22 nm) than at pH 7.5 (about 23 nm). However, an additional pH-shift experiment revealed that this change in diameter is not related to the pH change but is rather a consequence of the range of diameters present in the different capsid samples prepared for the pH 7.5 and pH 5.5 experiments (see Figs. 2a, d, S12, S13, S14, S15).

Secondary structure elements derived by NMR and obtained from the cryo-EM structure of ACNDV Cp at pH 7.5 agree well (Fig. 1). There seems to be an exception for residues 14–19 that according to NMR form a β-strand. This is consistent with the characteristic β-strand zigzag pattern in the ACNDV Cp structure (Fig. S11a), though, the backbone conformation deviates from ideal β-strand geometry

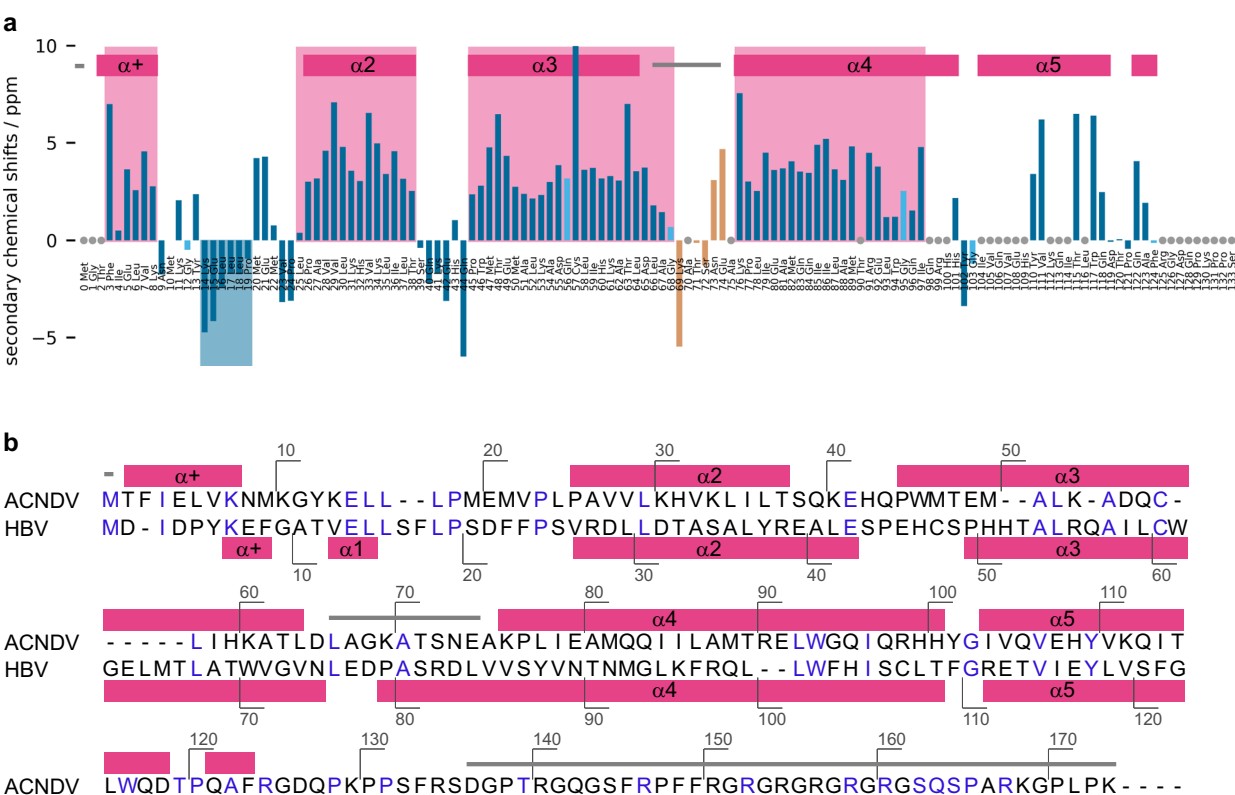

**Fig. 1 | ACNDV Cp secondary structure determined by solid-state NMR and alignment of the ACNDV Cp sequence with HBV Cp. a** Secondary chemical shifts of ACNDV Cp at pH 7.5 calculated from assigned CA and CB resonances. Pink background highlights where ≥4 positive secondary chemicals shifts appear in row (α-helix) and blue background where ≥3 negative secondary chemical shifts appear in a row (β-strand). Light blue vertical lines stand for secondary chemical shifts for which only the CA chemical shift was considered either because the residue is a glycine or because the CB chemical shift could not be assigned. Orange lines indicate tentative assignments (Fig. 4) and gray circles represent unassigned residues. Dark pink horizontal bars show α-helices from the cryo-EM structure of ACNDV chain A and gray horizontal lines indicate regions where the map density was not resolved. **b** The alignment of ACNDV Cp with HBV Cp was taken and adapted from Lauber et al.[1]. Annotations for secondary structure elements of ACNDV Cp pH 7.5 chain A are shown above the sequence (DSSP algorithm[44] in ChimeraX). Gray lines indicate missing residues due to unresolved map density. Helix annotations of HBV Cp chain A of the HBV T = 3 Cp (pdb 6ui6) from[29] are shown below the HBV Cp sequence. A small helix, here termed α + as in ACNDV Cp, spans residues 7–9. Note 1.) Our ACNDV plasmid construct used for protein production contained an additional glycine between the first two residues methionine and threonine. Note 2.) The HBV genotype in this adapted Figure is hepatitis B virus genotype D subtype ayw (isolate France/Tiollais/1979) (HBV-D)[1] and the genotype of HBV T = 3 capsids (pdb 6ui6[29]) used as reference structure throughout this paper is Hepatitis B virus genotype A2 subtype adw (isolate Japan/Nishioka/1983) (HBV-A). Considering residues 1-149, the two protein sequences differ in positions 74 (N/V), 87 (N/S), 97 (I/F), and 116 (L/I). In addition, all cysteines were mutated to alanines in ref. [29] (pdb 6ui6).

enough that it is not recognized as a strand by DSSP in ChimeraX[44]. Residues 66-74 were omitted in the structure because the cryo-EM map density was not well resolved at the tip of the spike for pH 7.5 and 5.5 capsids (Figs. S6, S7, and S8), yet the connection between the spike helices was visible in local resolution filtered maps (Fig. S17). It was possible to tentatively assign additional NMR peaks to the ACNDV Cp spike loop region (Fig. 4). Even though some of these peaks are well resolved and of reasonable intensity−e.g., 73N in Fig. 4b−no cross peaks to neighboring residues were found in 3D NMR spectra at the given signal-to-noise ratio, probably due to poor polarization transfer as a consequence of dynamical effects. The assignment of these residues is based on excluding other options for a certain amino-acid type (Table S5). Positive secondary chemical shifts between 66L and 68G suggest that the helical character is maintained until residue 68 (Fig. 1a). From residue 69 K until 75A, the signs of the secondary chemical shifts do not show a clear pattern typical for a defined secondary structure.

Even though residues in helix α5 are well resolved by cryo-EM, many residues in helix α5 in a stretch between aa 104-118 (Fig. 1a) could not be assigned in NMR spectra. This is most likely due low intensity

caused by heterogeneous line broadening in this region, which is caused by an unresolved splitting of the resonances according to the three chains. Peak splitting may occur for quasi-equivalent protein chains with small structural differences due to local adaptation of the chains to the icosahedral symmetry as was previously observed with HBV Cp[22]. Indeed, the hand region is the protein stretch for which the largest differences between ACNDV AB and CC dimers were observed in the cryo-EM structure (Fig. 3b). In the N-terminal part no peaks are missing except for the very first three residues (vide supra). Furthermore, for the 30 C-terminal residues (140 onwards), no peaks are visible in the NMR spectra. The C-terminus is thus flexible on an intermediate time scale similar to the situation in HBV capsids[45].

In contrast to the HBV capsids[22] no clear signs for extensive peak splitting were observed in the NMR spectra of ACNDV capsids, with the exception of 3F, which shows multiple peaks in the hNH spectrum (Fig. S3a), possibly due to a different conformation of the three 3F amide protons in the asymmetric unit. Peak broadening observed for several peaks may point to further splittings, which are however not resolved, as listed in Table S6 and shown on the structure in Fig. S13. Peak broadening was indeed observed for all three tryptophans: 46W,

**Table 1 | Cryo-EM data collection, refinement and validation statistics**

| | ACNDV pH 7.5 (EMDB-15295) (PDB 8AAC) | ACNDV pH 5.5 (EMDB-16371) (PDB 8COO) | ACNDV pH 7.5 | ACNDV pH 5.5 | ACNDV pH 5.5 shifted to pH 7.5 |
|---|---|---|---|---|---|
| **Data collection and processing** | | | | | |
| Magnification | 130,000 | 166,600 | 129,000 | 129,000 | 129,000 |
| Voltage (kV) | 300 | 300 | 300 | 300 | 300 |
| Electron exposure (e–/Å²) | 76 | 55 | 77 | 77 | 77 |
| Defocus range (μm) | –0.5 to –2 | –0.2 to –2 | –1 to –2.6 | –1 to –2.6 | –1 to –2.6 |
| Pixel size (Å) | 1.07 | 0.845 | 1.087 | 1.087 | 1.087 |
| Symmetry imposed | I | I | I | I | I |
| Initial particle images (no.) | 375,654 | 215,225 | 146,077 | 346,287 | 257,179 |
| Final particle images (no.) | 70,868 | 77,129 | 38,355 | 141,485 | 61,336 |
| Map resolution (Å) | 3.7 | 3.9 | 4.1 | 3.8 | 3.9 |
| FSC threshold | 0.143 | 0.143 | 0.143 | 0.143 | 0.143 |
| Map resolution range (Å) | 3.2–8.0 | 3.4–7.9 | 3.4–6.3 | 3.4–6.7 | 3.4–6.4 |
| **Refinement** | | | | | |
| Initial model used (PDB code) | de novo | de novo | | | |
| Model resolution (Å) | 3.9 | 4.2[a] | | | |
| FSC threshold | 0.5 | 0.5 | | | |
| Map sharpening $B$ factor (Å²) | –190 | –280 | | | |
| Model composition | | | | | |
| Non-hydrogen atoms | 198,540 (1103 per chain) | 197,460 (1097 per chain) | | | |
| Protein residues | 137 per chain | 136 per chain | | | |
| $B$ factors (Å²) | | | | | |
| Protein | 68.73 | 55.67 | | | |
| R.m.s. deviations | | | | | |
| Bond lengths (Å) | 0.008 | 0.005 | | | |
| Bond angles (°) | 1.06 | 1.144 | | | |
| Validation | | | | | |
| MolProbity score | 0.89 | 1.24 | | | |
| Clashscore | 1.47 | 2.04 | | | |
| Poor rotamers (%) | 0 | 0.9 | | | |
| Ramachandran plot | | | | | |
| Favored (%) | 98.02 | 96.05 | | | |
| Allowed (%) | 1.98 | 3.95 | | | |
| Disallowed (%) | 0 | 0 | | | |

[a]The model resolution is given for the stable core (residues 2–65, 75–135 chain A, 2–65, 75–134 chain B, 2–64, 76–132 chain C). When flexible parts at the tip of the spikes and at the N-termini are included in the overall map-model resolution estimate is 7.4 Å.

94W, and 117W, for which CB and CG resonances are broadened. In the hNH spectrum, at least six partially overlapping peaks for NE1-HE1 Trp sidechains were counted (Fig. S3a). With only three tryptophans in the protein sequence, this suggests that at least two out of the three tryptophans have different HE1 chemical shifts for the chains in the asymmetric unit. The three tryptophans are located in different parts of the protein sequence: 46 W at the base of helix α3, 94 W in the lower part of helix α4, and 117 W at the end of helix α5. 46 W and 94 W lie in protein segments that are almost identical in the three chains of the asymmetric unit. It appears that the tryptophan sidechain resonances are particularly sensitive to the local environment even if geometric changes are very small. Other residues for which peak broadening was observed lie at the N-terminus (3F, 4I), in the region after helix α + (14 K), in a cluster at the beginning of helix α2 and in the close-by helix α4 (26 P, 29 V, 31K, 94W, 97I), in the middle of helix α5 (111 V), and in the hand region (117W, 120T, 123A) (Fig. S16).

Unassigned cryo-EM map density located below the spikes of AB dimers but not below CC dimers was found in the ACNDV capsid maps (Fig. S17). Likely sources of the diffuse density below AB dimers are enclosed nucleotides and/or the flexible CTD of the capsid protein.

## Discussion

We successfully prepared ACNDV capsids by recombinant expression in *E. coli* and determined the 3D molecular structure by cryo-EM at a resolution of 3.7 Å. This high-resolution structural model of a nackednaviral capsid enables first direct comparison with the core proteins and capsids of the hepadnaviruses.

The evolutionary lineages of hepadna- and nackednaviruses separated >400 million years ago. Development of T = 4 symmetry and of the envelope in hepadnaviruses occurred after this separation event[1]. Intriguingly, despite these differences, the overall 3D fold, including most secondary-structure elements, of the capsid-forming core proteins have remained essentially identical in the two virus families (Fig. 5a, b). In both HBV and ACNDV, two Cp monomers associate into stable dimers via two central helices, the resulting four-helix bundles generate outward pointing spikes, and the N-termini up to the beginning of helix α3 are wrapped around the spike helices; helix α5 together with subsequent residues forms a hand-like region providing the essential interdimer contacts. The fold of the DHBV Cp, the type member of the *avihepadnaviruses*, is also similar to ACNDV Cp, with the main difference of an additional extension domain at the DHBV capsid spike (Fig. 5c).

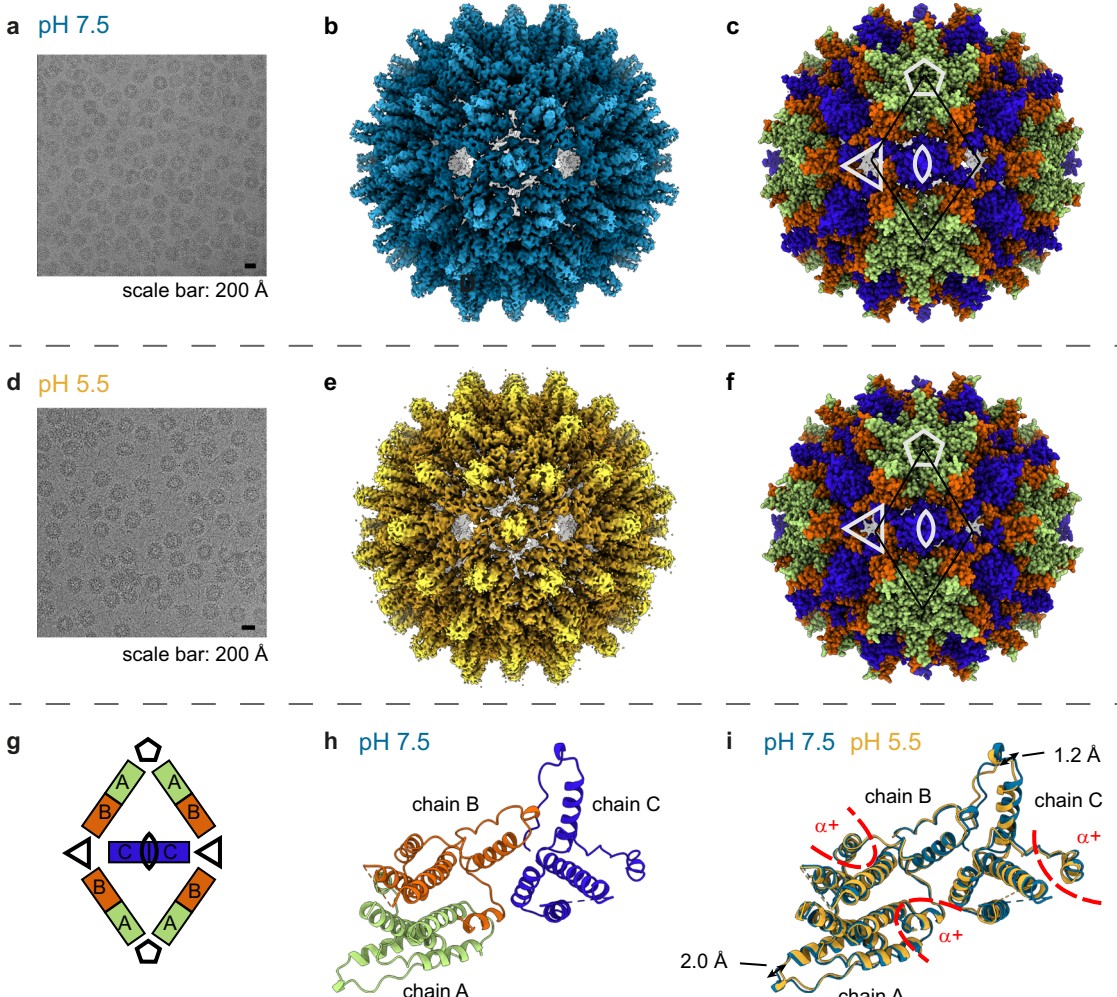

**Fig. 2 | ACNDV Cp at pH 7.5 and pH 5.5 have the same protein fold. a**, **d** Cryo-EM micrograph of ACNDV capsids at pH 7.5 and pH 5.5 showing icosahedral $T = 3$ particles. **b**, **e** Cryo-EM density maps of the ACNDV capsid at pH 7.5, (resolution of 3.7 Å) plotted at a contour level of 4σ and of the ACNDV capsid at pH 5.5 (resolution of 3.9 Å plotted at a contour level of 3σ. **c**, **f** Whole capsid structures where chain A of the asymmetric unit is colored in green, chain B in orange and chain C in blue. The white shapes indicate the symmetry axes in the $T = 3$ particle. Chains within the black rhombus are schematically shown in panel **g**. **g** Schematic representation of chains showing their organization in $T = 3$ symmetry capsids. **h** Asymmetric unit of the ACNDV capsid at pH 7.5. The spike residues 66–74 and the N-terminal residue were omitted in the structure due to a lack of well-resolved map density. A dotted line is displayed to show the connectivity of the protein at the spikes. **i** Superposition (matchmaker in ChimeraX[80]) of the asymmetric units of the ACNDV capsid at pH 7.5 (blue) and pH 5.5 (yellow). The positions of the α + helices are marked in red.

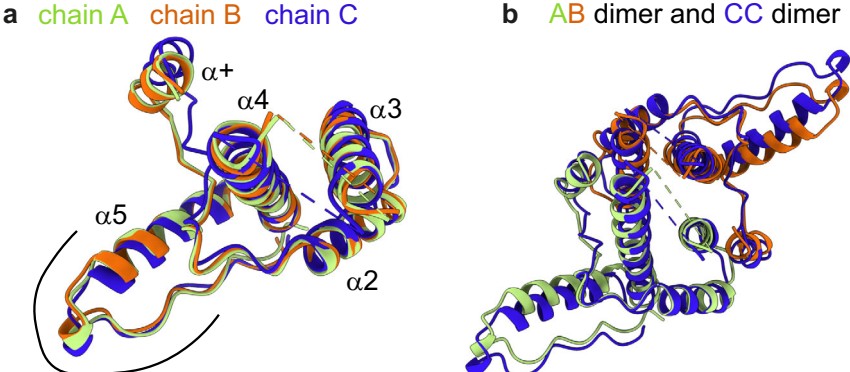

**Fig. 3 | ACNDV Cp chains at pH 7.5. a** Chain A, chain B, and chain C of ACNDV Cp at pH 7.5 superimposed by "matchmaker" implemented in ChimeraX[80]. The helix numbering convention is the same as in[1] and the hand region is delineated by a black line. **b** Superposition of the AB and CC dimers.

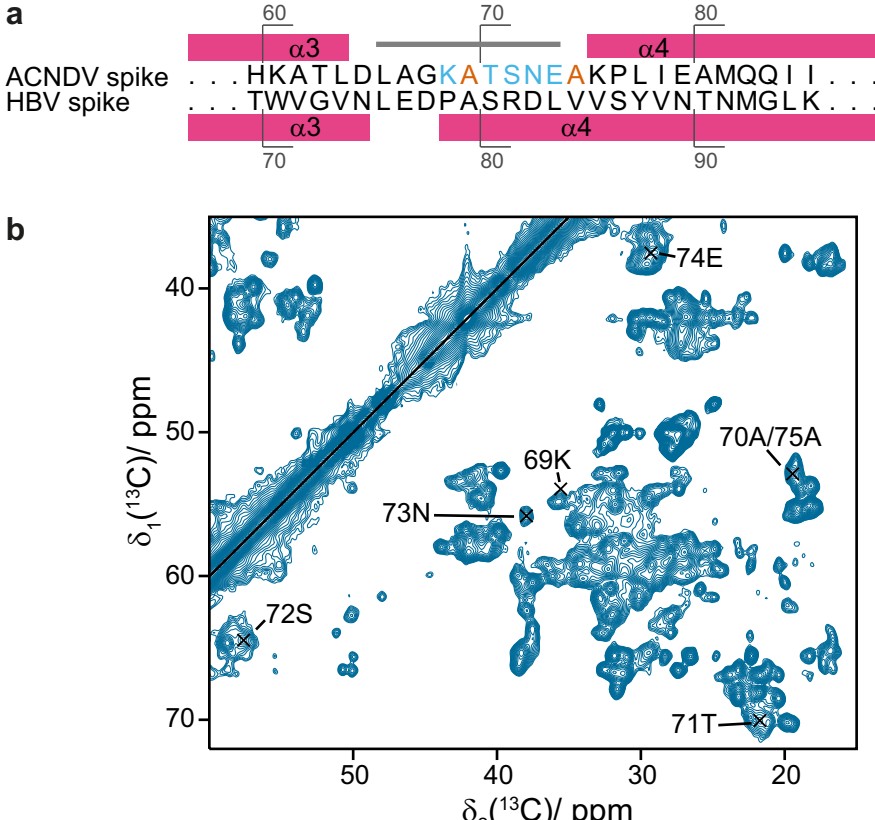

**Fig. 4 | Spike residues were only observed by NMR not by cryo-EM. a** Alignment of the sequences at the spike tips for ACNDV Cp (top row) and HBV Cp (bottom row). The alignment was adapted from[1]. Residues in the ACNDV sequence in light blue are the ones tentatively assigned by NMR (see panel **b**). The peak labeled 70 A/75 A in panel **b** could belong to either alanine residue (orange in panel **a**). Other ACNDV residues in black are unambiguously assigned (Table S5). $n = 1$ independent experiments have been recorded. **b** DARR spectrum (20 ms mixing time) of ACNDV capsids at pH 7.5 labeling the peaks that were tentatively assigned to spike residues. The assignment of these peaks was done via exclusion principle because sequential connections in the 3D assignment spectra were missing. The peak for 69K has only low intensity and is close to a stronger peak.

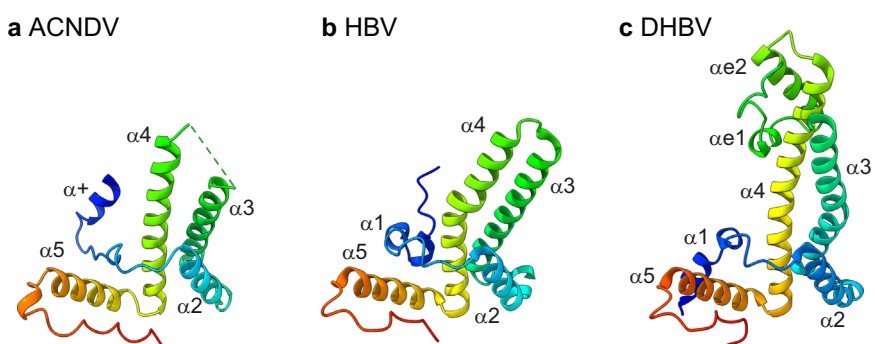

**Fig. 5 | Cp structures of ACNDV, HBV, and DHBV. a** Chain A ACNDV Cp at pH 7.5. The dotted line indicates where the non-resolved spike residues were omitted. **b** Chain A HBV Cp T = 3 (pdb 6ui6)[29]. **c** Chain A DHBV Cp T = 4 (pdb 6ygh)[20].

The obtained 3D structure largely corroborates the alignment of HBV and ACNDV Cp by Lauber et al.[1] (Fig. 1b). Most secondary-structure elements were predicted correctly based on the primary sequence in ref. [1] except that a β-strand was predicted in place of helix α5. Based on the alignment, ACNDV Cp may be divided into an N-terminal assembly domain (NTD), a C-terminal RNA binding domain (CTD), and a linker, as in HBV Cp. The homologous N-terminal domain in ACNDV Cp comprises residues 1–135, the linker 136–144, and the C-terminal domain 145–174. Indeed, ACNDV Cp truncated after residue 146 still forms intact $T = 3$ icosahedral particles as shown in ref. [1]. Based on cryo-EM, we can conclude that residues 136 onwards are flexible with 135 R (chains B and C) or 136 S (chain A) being the last resolved residues. By comparison, HBV Cp consists of an NTD comprising residues 1–140, the linker spans residues 141–149, and the CTD is made up of residues 150 onwards. The resolved part of the NTD in HBV Cp149 $T = 3$ comprises 142 or 143 residues, depending on the quasi-equivalent chain[29], which is the full NTD plus two or three residues of the linker, similar as in ACNDV Cp where the presumed NTD (1–135) is resolved but not the presumed linker region (136–144) and CTD (145–174).

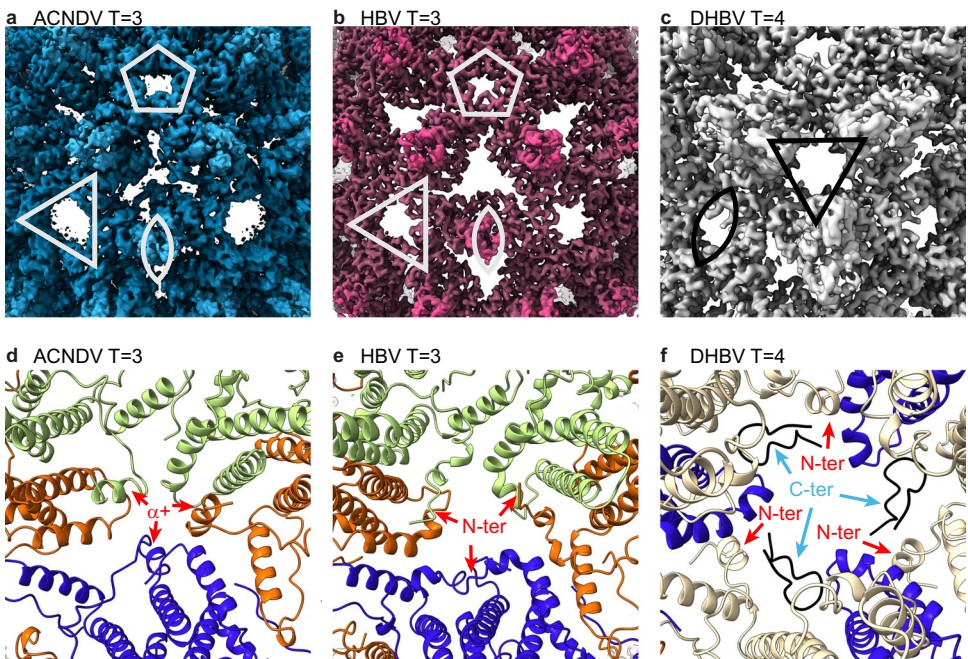

**Fig. 6 | The N-terminus is different for ACNDV, HBV, and DHBV Cp. a** Cryo-EM density map (3.7 Å) of the ACNDV capsid at pH 7.5 plotted at a contour level of 4σ. The center of the image shows the region between the fivefold, threefold, and twofold axis where the additional α + helices partially occlude the triangular holes. Symmetry axes are marked with white symbols. **b** Cryo-EM density map of the *T* = 3 HBV capsid from[29] (emd-20669) with a resolution of 3.5 Å plotted at a contour level of 4σ. The hole at the local threefold axis in the middle of the image is larger because HBV Cp lacks the α + helix. **c** Cryo-EM density map of the DHBV *T* = 4 capsid (emd−10800) with a resolution of 3.7 Å[20] plotted at a contour level of

3σ. A zoom onto the threefold axis is shown and symmetry elements are marked with black symbols. **d** ACNDV Cp pH 7.5 chain A (green) chain B (orange), and chain C (blue) showing how the α + helices (marked with red arrows) protrude in direction of the capsid surface. **e** HBV Cp chain A (green) chain B (orange), and chain C (blue) (pdb 6ui6). The direction of N-termini (marked with red arrows) is tangential to the capsid shell. **f** Structure (pdb 6ygh) of the DHBV capsid, with chain C (blue), chain D (beige), and C-termini (black). The N-termini point towards the capsid lumen (red arrows) while the C-termini emerge from the capsid inside (light blue arrows).

The nucleic-acid binding capacity of hepadnaviral capsids correlates with the number of internally accessible positively charged sidechains, largely found in the CTD[4,18,46]. For instance, both HBV and DHBV encoding Cp with truncated or mutated CTDs lacking part of the arginine residues do not form stable nucleocapsids harboring full-length rcDNA but rather shorter DNA derived from less-than-genome sized spliced pgRNAs[47-49]. Plausibly, the reduced number of arginine sidechains may be unable to electrostatically compensate for a largely double-stranded full-length DNA such that the capsid shells are disrupted early during second strand DNA synthesis; conversely, particles containing shorter reverse transcription products, such as from the major 2.0 kb SP1 splice product, would still be stable.

The HBV Cp CTD (aa 150–183) contains 16 arginines and one glutamic acid, yielding an excess of 15 positively charged residues within the CTD in order to compensate for the negatively charged nucleic acids. ACNDV Cp CTD (aa 145–174), on the other hand, contains only 8 arginines, two lysines, and neither glutamic acid nor aspartic acid, which is an excess of 10 positively charged residues. Accessible basic and acidic NTD residues within the inner lining of the capsid shell comprise 11K, 14K, 41K, 42E, 99R, 108E, and 112K for ACNDV Cp and 46E, 112R, 113E, and 117E for HBV Cp (Fig. S18). There are three more positively charged than negatively charged amino acids within the accessible ACNDV Cp NTD, and two more negative charges than positive ones for HBV Cp and the electrostatic surface potentials show a more negatively charged inner surface for ACNDV than HBV capsids (Fig. S17). If accessible amino acids within the NTD are added to charges within the CTD, there is a comparable excess of positive charge for ACNDV Cp vs. HBV Cp. We observe that the RNA shell in radial profiles of cryo-EM reconstructions of the ACNDV capsid is located at a distance of about 2.8 nm to the capsid walls, which is similar to the distance observed for HBV virus of about 3 nm[25]

(Fig. S19). Still, the packaging capacity would be smaller for ACNDV Cp because it exclusively assembles into *T* = 3 capsids which means that 25% less protein is available for neutralization of nucleic acid phosphates. To date, the charge compensation mechanisms of nucleic acids in nackednaviruses is largely unknown. Incorporation of additional charges such as cations may help to add to the charge compensation of ACNDV capsids in vivo. Since genomes of HBV and nackednaviruses are of comparable size of around 3000 bp, the genomes of nackednaviruses appear to require additional charge compensation mechanisms to form stable nucleocapsids.

The most distinguishing feature of the ACNDV Cp structure is the α + helix (Fig. 6), which was already apparent in the lower-resolution cryo-EM map (emd-3822)[1]. A major difference between ACNDV and HBV capsids is the N-terminus, which is directed towards the capsid outside in ACNDV Cp, while in HBV Cp the direction is tangential to the capsid shell (Fig. 6d, e). Comparing ACNDV and HBV capsids at the level of quaternary structure, the larger size of the fenestrae in ACNDV capsids at the fivefold and the threefold axes is remarkable (Fig. 6a, b). In addition, the triangular holes between the symmetry axes are partially closed by the α + helices in ACNDV, whereas large holes are visible for HBV capsids. Also, the smaller opening of the ACNDV capsid triangular hole (in the center of Fig. 6a) is compensated by bigger openings at the threefold axes (marked with a triangle in Fig. 6a). Analysis with the HOLE software[50] yields a hole radius of 1.1 Å for the local threefold axis of the ACNDV capsid at the narrowest point, while the equivalent holes measure 4.8 Å for the HBV and 6.1 Å for DHBV capsid. The 3-fold axis, on the other hand, takes a minimum radius of 6.1 Å in the ACNDV capsid and 4.8 Å in HBV; the two-fold axis in DHBV has a minimum radius of 3.3 Å. The hydrophobic/hydrophilic outer surfaces of ACNDV, HBV, and DHBV Cp show a similar pattern (Fig. S20). A difference can be seen around residues 25L and 87L

(ACNDV numbering) where ACNDV Cp seems to be slightly more hydrophobic than HBV Cp; in DHBV Cp this region is partly occluded by the extension domain. The outer surface of ACNDV is more positively charged than HBV and DHBV capsids (Figs. S18, S20) similar as at the capsid interior (Fig. S18). In conclusion, diffusion of small molecules through the ACNDV capsid will hardly be limited by the occlusion of triangular pores by the α + helix. This finding is quite unexpected for an unenveloped virus which could support that ACNDV may adapt a quasi-envelope without actually encoding for the corresponding surface protein, similar to what has been described in hepatitis A viruses[13,14] and hepatitis E viruses[15]. Such a quasi-envelope would protect the genetic material from outside influences.

In HBV, the large triangular openings might be important for exposing the CTD during certain steps during the viral life cycle[18,51], e.g., import of the capsid through the nuclear pore, the CTD has to protrude out of the capsid shell in order to interact with importins[52].

Interestingly, in DHBV capsids, the positions where the α + helices of ACNDV lie, are occupied by the very C-termini which emerge to the capsid shell (Fig. 6c, f)[20]. It was speculated that the outward peeking C-terminus of DHBV could be part of a capsid externalization mechanism although no such protrusion was experimentally observed[20]. The N-terminus of DHBV Cp directs away from helices α2 and α3 of the opposing monomer, towards the inside of the particle. In a dimer, one N-terminus is directed towards the spike base while the other points underneath the hand region.

For ACNDV capsids, an unfolding and subsequent externalization of the N-terminus would be possible since it already points towards the capsid exterior. We however did not observe a state with outwards-peeking N-termini in the cryo-EM studies. We speculated that at pH 5.5 the ACNDV capsid would be less stable and an intermediate state with externalized N-termini could be visible. Indeed, the capsids sedimented in the NMR rotor denatured quicker at pH 5.5, which may indicate that they are less stable (Fig. S3c) and at pH 4.0 no intact capsids were visible by TEM (Fig. S21) but a state with more outwards peeking α + helices was not observed. In addition, an unusually high downfield chemical shift of the 3F amide hydrogen was detected in the NMR hNH spectrum (Fig. S4). The reason is not clear and a strong hydrogen bond that could potentially cause such a shift is not visible in the cryo-EM map.

Although the arrangement of spike helices is different in ACNDV Cp compared to HBV Cp, the difference is not as pronounced as for DHBV Cp vs. HBV Cp: the direction of ACNDV Cp spike helices is intermediate between HBV Cp and DHBV Cp (Fig. 5). DHBV spikes contain an extension domain with two additional helices (αe1 and αe2)[20]. The HBV spike seems to be mainly stabilized by hydrophobic interactions, for example by 88Y-71W contacts of facing chains via π-π-interactions of the aromatic sidechains (Fig. S11f) and seems to be overall more hydrophobic at the spike interfaces than ACNDV Cp (Fig. S20). A cysteine bridge can be formed across the HBV Cp spikes as seen for example in the crystal structure of the HBV $T = 4$ capsid (pdb 1qgt) (Fig. S11e)[53]. However, the formation of the bond is not essential for capsid formation[54]. The cysteine is conserved in ACNDV Cp (57 C) as illustrated in the alignment in Fig. 1b. However, the two 57 C in the ACNDV Cp dimer are too far apart to form a disulfide bond (Fig. S11d). While protein expression and purification in absence of dithiothreitol (DTT) would allow for disulfide bond formation, the latter was never observed, neither in non-reducing SDS-PAGE nor by investigating the characteristic cysteine chemical shift in NMR spectra (Fig. S23), clearly demonstrating that a cysteine bond is not formed in ACNDV capsids. A cysteine bond is also missing in DHBV capsids, where the spike cysteine is not conserved and interface interactions comprise 52H-52H and 60F-60F of opposing monomers across the spikes (Fig. S11g)[20].

Three core motifs: motif I (between helix α1 and α2), motif II (at the bottom of helix α3), and motif III (hand region) are known to be conserved in the amino-acid sequences of *Hepadnaviridae*[55]. Comparison of the motifs in the capsid structures of HBV, DHBV, and ACNDV (Fig. S11h–j) shows that these three regions are structurally conserved among the representative members of hepadnaviruses (HBV and DHBV) and nackednaviruses (ACNDV), which is in line with the sequence alignment in ref. [1]. The inter-dimer contacts (motif I and III) thus seem to be particularly conserved, while the C-termini, N-termini, spike tips, and intra-dimer contacts such as cysteine bonds are neither conserved among hepadnaviruses nor with nackednaviruses.

The capsid protein conformation has to adapt for icosahedral symmetry with three or four quasi-equivalent chains within the asymmetric unit of $T = 3$ and $T = 4$ capsids, respectively. Indeed, in NMR studies of HBV $T = 4$ capsids, peak splitting marking such adaptation was observed in particular between residues 10–20 (around helix α1) and 120–140 (hand region)[22]. Also, HBV Cp140 $T = 3$ capsids displayed the peak-splitting pattern indicative for local symmetry adaptation[56]. The hand region seems to be involved in symmetry adaptation for all of the four capsids that we looked at: ACNDV capsids at different pH (Fig. 3b and Fig. S11b), HBV $T = 3$, HBV $T = 4$, and DHBV capsids (Fig. S20), whereas the hand region orientation is slightly different when comparing the protein chains. The hand region is also one of the most conserved parts of nackedna- and hepadna viruses (see alignment in[1] and Fig. S11B and C), indicating its importance in capsid assembly and symmetry adaptation. We therefore expected to observe NMR peak splitting within the hand region of ACNDV Cp and potentially also within the loop region after helix α+. However, only vague evidence of peak splitting was found (Table S6 and Fig. S13), except for residue 3f and three tryptophan sidechains (46 W, 94 W, and 117 W). Most importantly, no peak splitting for backbone atoms was observed. Even though the hand region of ACNDV Cp is well resolved in cryo-EM, many residues could not be assigned by NMR (Fig. 1a). A possible explanation is peak broadening, either as a non-resolved peak splitting[22] or from stochastic disorder. The lack of evident peak splitting, but the presence of peak broadening suggests that the subunits within the ACNDV icosahedral capsid are statically more disordered than in HBV, also supported by the improved cryo-EM map resolution upon symmetry expansion and refinement in C1 symmetry (Fig. S8).

When compared to 2D DARR spectra of HBV capsids, the peaks in the spectra of ACNDV capsids have larger NMR line widths. While for HBV capsids full widths at half maximum (FWHM) between 70 and 100 Hz for HBV capsids[45] have been reported, for ACNDV, FWHMs of 18 isolated peaks lie between 100–171 Hz, with an average of 127 Hz. In proton-detected hNH spectra of ACNDV capsids, the line widths of five isolated peaks (12G, 14K, 17L, 38T, and 44Q) lie between 102 and 163 Hz, with an average of 139 Hz (no apodization). For HBV capsids, the average FWHM of all peaks of a fully protonated sample was 170 Hz for the $^1$H dimension (no apodization) as reported in ref. [57]. For a fair comparison ACNDV capsid line widths should be compared to the isolated peaks in ref. [57], which often have widths below 100 Hz. Peaks in the hNH spectrum of ACNDV capsids (Fig. S4a) are thus most likely less well resolved than for HBV capsids[57]. The comparison of $T_2'$ 1D hNH bulk relaxation times shows that fully protonated ACNDV and HBV capsids have similar contributions to the homogeneous linewidth ($T_2' = 2.3$ ms for ACNDV and 2.2 ms for HBV, see Table S7). It is also interesting that carbon-detected measurements at higher magnetic field (proton resonance frequency of 1200 MHz) yielded improved resolution for HBV but less so for ACNDV capsid samples[58]. These findings imply that heterogeneous broadening and thus structural heterogeneity, indicated by non-resolved peak splitting of the individual monomers, is generally larger for ACNDV than for HBV capsids.

In addition, the spike tips were less well resolved in ACNDV than in HBV capsids (Figs. S6, S7, S8 and S17) although, also in HBV capsids cryo-EM map resolution decreases in this region[29]. The loop connecting the spike helices comprises only three residues in HBV (Fig. 1b) but about seven in ACNDV capsids, as shown by NMR secondary chemical shifts (Figs. 1 and 4). This may allow for larger dynamical disorder in the NMR spectra and impair resolution in the cryo-EM map.

Several structural elements in HBV capsids have been implied to mediate $T = 3/T = 4$ dimorphism. Wu et al.[29] suggested the DFFP motif (residues 22–25 and part of the conserved motif I shown in Fig. S10h–j) to be responsible for the formation of $T = 4$ capsids since this motif is highly conserved within the hepadnaviruses but not in nackednaviruses where $T = 3/T = 4$ dimorphism has not been observed. In ACNDV Cp the DFFP motif is replaced by EMVP (residues 21–24) (Fig. 1b)[29], which yields a very similar backbone trace to DFFP in HBV Cp and DHBV Cp (Fig. S21). In HBV and DHBV capsids, the region around DFFP connects the segment upstream of helix α2 with the hand region, largely by hydrophobic sidechain interactions. The smaller M22/V23 sidechains in ACNDV Cp compared to F23/F24 in HBV Cp may decrease the strength of these interactions although it remains unclear how this would affect $T = 3/T = 4$ dimorphism. It was also proposed[29] that the additional helix α+ of ACNDV Cp resembles the α-helical N-terminus of the hepatitis B e-antigen (HBeAg) of HBV[59], an N-terminally extended but CTD-less non-assembling version of Cp[4]. A recombinant HBeAg with the identical sequence as HBV Cp149 but carrying the 10 residue N-terminal extension (HBV Cp(−10)149) formed mainly $T = 3$ particles under non-physiological conditions[60], hence the presence of an extra N-terminal α-helix may favor $T = 3$ over $T = 4$ symmetry[29]. More firmly established as influencing HBV capsid morphogenesis is the linker (residues 141–149) connecting NTD and CTD. HBV Cp truncated at residue 140 (Cp140) assembles predominantly into $T = 3$ capsids[27,28] although assembly was overall less efficient than for Cp149. This led to the assumption that Cp140 capsids are more fragile and aggregate more easily than Cp149 capsids[28]. Nevertheless, NMR spectra of Cp140 are well resolved and the characteristic peak splitting at the hinge regions is $T = 3$ specific but otherwise similar as in Cp149[56]. In ACNDV Cp the residues downstream of position 135 are highly flexible, suggesting that truncation up to this point (which comprises the whole NTD) should still be compatible with capsid assembly. ACNDV Cp 146 indeed forms particles[1] but if and how the putative linker sequence downstream residue 135 affects capsid symmetry remains to be established.

Based on a protocol for HBV Cp[45], preparation of ACNDV Cp also involved the use the detergent Triton-X-100, which was recently shown to bind into a hydrophobic pocket between the spike helices at the HBV Cp dimer interface[56]. Indeed, we found unaccounted EM map density at the base of AB ACNDV Cp dimers (Fig. S17). However, the hydrophobic surface representation shows that ACNDV Cp lacks a large hydrophobic interface between adjacent monomers as in HBV Cp (Figs. S20, S22), hence the unassigned density more likely corresponds to bound RNA or the flexible CTDs. While still hypothetical, binding of a natural pocket factor into the HBV Cp hydrophobic pocket may be related to the capsid maturation and envelopment[56,61,62]. However, compared to HBV, DHBV Cp does not feature an equally large hydrophobic area at the dimer interface but displays two small hydrophobic patches shown in Fig. S22: one at the extension domain and one towards the spike base.

We here solve, using cryo-EM supported by solid-state NMR, the high-resolution structure of ACNDV capsids which allows a structural comparison of the two related but distinct families of nackednaviruses and hepadnaviruses. Even if both lineages evolutionarily separated >400 million years ago[1] and only hepadnaviruses acquired an envelope, the overall capsid protein fold and capsid architectures remained the same. Hence, although we expected that envelopment implies fundamental differences in virus life-style, including host and tissue tropism, its acquisition is found to have only a minor effect on the capsid protein fold.

Nonetheless, distinct differences between ACNDV and HBV capsids are revealed, including a substantially lower positive charge in the interior of the $T = 3$ ACNDV capsids. Given the similar genome sizes of about 3 kb of both nackedna- and hepadnaviruses this implies that additional positive charges may be taken up into the capsid to compensate for the negatively charged nucleic acids. The ACNDV capsids

produced in vitro were exclusively $T = 3$ while HBV exists in both $T = 3$ and $T = 4$ with $T = 4$ being the main species. In general, the assemble ACNDV viral particles are found to be less resistant than HBV against denaturation with time in storage and low pH as detected by NMR, as well as with exposure to GuHCl.

## Methods

### Sample preparation of African cichlid nackednavirus capsid protein

ACNDV Cp was expressed in *E. coli* and spontaneously assembled particles were purified by sedimentation in sucrose gradients in TRIS-HCl buffer at pH 7.5 similar as in refs. [34,45], and ref. [19]. To assess pH dependent alterations, purified ACNDV capsids were dialyzed against acetate buffer, pH 5.5. All samples were sedimented into magic-angle spinning (MAS) rotors[35,36] using an ultracentrifuge and home-built filling-tools[63].

For expression of the ACNDV Cp, the HBV Cp coding sequence in plasmid pRSF-T7-HBcopt[18] was replaced by a synthetic DNA string encoding the 174 aa ACNDV Cp sequence (Genbank ID: GenBank: AZP02123.1) with an extra glycine residue immediately following the start Met residue (for cloning reasons) using an *E. coli* codon usage adapted nucleotide sequence (GeneOptimizer software; ThermoFisher / GeneArt). Main features of the resulting pRSF-T7-CNDVcOpt plasmid are an RSF origin, a kanamycin resistance gene, and a bacteriophage T7 promoter controlling ACNDV Cp expression in *E. coli* strains providing T7 RNA polymerase, such as BL21(DE3) and derivatives.

Expression of ACNDV capsid protein in 1 L of $^{15}$N-$^{13}$C labeled medium yielded enough sample to fill a 3.2 mm NMR rotor with about 35 mg of protein. For some samples, an additional purification step by size exclusion chromatography was added but this did not improve sample quality, as assessed by 2D NMR spectroscopy. Purified protein samples were analyzed by SDS-PAGE, transmission electron microscopy (TEM), and 2D carbon-carbon correlation spectra. Particular methods for specific samples: (a) Uniformly $^{13}$C-$^{15}$N labeled ACNDV capsids at pH 7.5 for NMR measurements: BL21*Codonplus cells transformed with expression plasmids for African cichlid nackednavirus capsid protein (pRSF-T7-CNDVcOpt) were grown overnight on agar plates containing chloramphenicol (35 mg/L) and kanamycin (50 mg/L) at 37 °C. For pre-cultures, single colonies were transferred into 15 mL tubes containing 5 mL LB, chloramphenicol (35 mg/mL), and kanamycin (50 mg/mL). Pre-cultures were grown at 250 rpm at 37 °C for about 6 h and transferred to 1 L of M9 medium containing $^{13}$C glucose and $^{15}$N ammonium chloride. The M9 culture was grown at 110 rpm at 37 °C for about 5 h until an OD$_{590}$ of 0.6 was reached. The temperature was lowered to 20 °C and overnight expression was induced with 1 mM isopropyl β-D-1-thiogalactopyranoside (IPTG). The cells were harvested by centrifugation for 25 min at 4'000 g and 4 °C. The pellet was resuspended in lysis buffer (50 mM tris-(hydroxymethyl)-aminomethan (TRIS), 50 mM NaCl, 5 mM ethylenediaminetetraacetic acid (EDTA), 5 mM dithiothreitol (DTT), pH adjusted to 7.5). Chicken egg white lysozyme (1 mg/mL), one tablet of protease inhibitor and Triton-X-100 (1% v/v) were added and the suspension stored on ice for 30 min. Benzonase nuclease was added and the sample kept at room temperature while rotating it. Cells were lysed by microfluidizing at 45 psi for 4 runs. The suspension was centrifuged for 30 min at 12,500 × $g$ and 4 °C. For purification, sucrose gradients from 10–60% w/v sucrose in lysis buffer were prepared in a SW32 swinging-bucket ultracentrifuge rotor. The sample was added on top of the sucrose solution and centrifugation was carried out for 3 h at 141,000 × $g$ and 4 °C: Fractions were harvested from top to bottom and stored at 4 °C. Most homogeneous fractions were selected based on SDS-PAGE and negative stain transmission electron microscopy (TEM) as shown in Fig. S1a–c. Selected fractions were concentrated in a 100 kDa cutoff concentrator and washed with HEPES buffer (50 mM 4-(2-hydroxyethyl)−1-piperazineethanesulfonic acid (HEPES), 5 mM DTT,

pH adjusted to 7.5) until the sucrose content was only about 0.3%. Filled NMR rotors were stored at 4 °C. Resulting data is shown in Figs. S1a–c and S3a. (b) Low pH screening: As additional purification step size exclusion chromatography was performed with a Sephacryl S400 resin. The column was equilibrated with TRIS lysis buffer (50 mM TRIS, 50 mM NaCl, 5 mM EDTA, 5 mM DTT, pH adjusted to 7.5). Fractions of the sucrose gradient purification containing protein were concentrated in a 100 kDa cutoff concentrator to a volume of 6–7 mL. After injection, 9–13 mL fractions were collected from the column at a speed of 1.3 mL/min. Protein samples were evaluated by TEM after size exclusion chromatography. Most homogeneous looking fractions were concentrated whereby the TRIS buffer at pH 7.5 was exchanged in a 100 kDa cutoff concentrator to K-phosphate pH 7.5, K-phosphate pH 5.8, acetate pH 5.5, acetate pH 5.0, or acetate pH 4.0 in a stepwise manner. Each of the buffers contained 50 mM buffer base+acid and 50 mM NaCl. Inspection of the protein samples by TEM revealed that samples in K-phosphate pH 7.5, K-phosphate pH 5.8 appear no different from the TRIS pH 7.5 sample. Protein samples in acetate buffer pH 4.0 contained aggregated material exclusively, while the sample in acetate buffer pH 5.0 contained single capsid particles and many aggregates. Acetate buffer pH 5.5 lead to many areas with nicely separated particles and only few areas with aggregates (Fig. S17). (c) Uniformly $^{13}C$-$^{15}N$ labeled ACNDV capsids at pH 5.5 for NMR measurements: After sucrose gradient centrifugation selected fractions were dialyzed (6–8 kDa cutoff) into acetate buffer at pH 5.5 (50 mM NaAc + acetic acid until pH 5.5, 50 mM NaCl) overnight at 4 °C. The sample was concentrated in a 100 kDa cutoff concentrator and filled in an NMR rotor of 3.2 mm diameter. Results are shown in Figs. S1d–f and S3c and (d) Uniformly $^{13}C$-$^{15}N$ labeled ACNDV capsids at pH 9.0 for NMR measurements: The work-up of the protein suspension was then performed in lysis buffer that contained no DTT. After sucrose gradient centrifugation the sample was stepwise subjected to TRIS buffer at pH 9.0 (50 mM TRIS base + TRIS HCl, 50 mM NaCl, pH adjusted to 9.0 with 1 M HCl) in a 100 kDa cutoff concentrator. Spectra are shown in Fig. S1g and H and Fig. S2b. (e) Uniformly $^{13}C$-$^{15}N$ labeled ACNDV capsids at pH 7.5 with missing lysine peaks: Protein samples were expressed and purified as described above in "(a) Uniformly $^{13}C$-$^{15}N$ labeled ACNDV capsids at pH 7.5 for NMR measurements". Size exclusion chromatography was performed with a Sephacryl S400 resin after sucrose gradient purification. The column was run with TRIS lysis buffer (50 mM TRIS, 50 mM NaCl, 5 mM EDTA, 5 mM DTT, pH adjusted to 7.5). Spectra are displayed in Figure S22. (f) Samples where (no) denaturation was observed (NMR and TEM). The protein sample of "(a) Uniformly $^{13}C$-$^{15}N$ labeled ACNDV capsids at pH 7.5 for NMR measurements" was repeatedly measured by 2D NMR spectroscopy. The sample stayed in the 3.2 mm NMR rotor during the whole time and was stored at 4 °C between measurements. The first measurement took place 2 months after NMR rotor filling. During the second measurement (5.5 months after sample filling) signs of denaturation were already visible in the NMR spectrum (data not shown). After almost 6.5 months clear signs of protein denaturation were detected (Fig. S4). (f) Uniformly $^{13}C$-$^{15}N$ labeled ACNDV capsids at pH 7.5 for proton-detection NMR measurements: Expression and purification took place as in "(a) Uniformly $^{13}C$-$^{15}N$ labeled ACNDV capsids at pH 7.5 for NMR measurements". After purification by sucrose gradient centrifugation a size exclusion chromatography (Sephacryl S400) step was added. The column was run with TRIS buffer (50 mM TRIS, 50 mM NaCl, 5 mM EDTA, 5 mM DTT, pH adjusted to 7.5). Collected fractions were selected for rotor filling based on homogeneity in TEM. The sample was concentrated in a 100 kDa cutoff concentrator and filled into an NMR rotor of 0.7 mm diameter. The hNH spectrum is shown in Fig. S3A. (g) Uniformly $^{13}C$-$^{15}N$ labeled ACNDV capsids sample without dithiothreitol (DTT): Plasmids (pRSF-T7-CNDVcOpt) were transformed into BL21 (DE3) CodonPlus RIL cells. Expression was carried out as in "(a) Uniformly $^{13}C$-$^{15}N$ labeled ACNDV capsids at pH 7.5 for NMR measurements" with the difference

that all buffers contained no DTT and that 1 mM PMSF was added to the lysis suspension. Protein was concentrated in a 100 kDa cutoff concentrator in TRIS buffer (50 mM TRIS, 50 mM NaCl, pH adjusted to 7.5) and filled into an NMR rotor of 3.2 mm diameter. An SDS-PAGE under non-reducing conditions was run (Fig. S19b) and a $^{13}C$–$^{13}C$ DARR spectrum recorded. No sign of dimerization that would indicate a disulfide bond was observed by SDS-PAGE. The cysteines were in reduced state as shown by NMR (Fig. S19a). (h) ACNDV capsid samples for cryo-EM measurements: Expression of ACNDV capsid protein sample that was used for cryo-EM measurements is described in "(b) Low pH screening". After size exclusion chromatography with pH 7.5 TRIS buffer (50 mM TRIS, 50 mM NaCl, 5 mM EDTA, 5 mM DTT, pH adjusted to 7.5) homogenous fractions were selected by TEM. The protein was concentrated in a 100 kDa cutoff concentrator and a suitable concentration for cryo-EM sample preparation was determined by TEM. For measurements at pH 5.5 the buffer was exchanged in a stepwise manner to acetate pH 5.5 buffer (50 mM sodium acetate + acetic acid until pH 5.5, 50 mM NaCl) in a 100 kDa cutoff concentrator. (i) Cryo-EM experiment for particle size determination: Plasmids (pRSF-T7-CNDVcOpt) were transformed into BL21 (DE3) CodonPlus RIL cells. Expression and purification was carried out as in "(a) with following differences: buffers contained no DTT, the cell pellet was frozen at −80 °C after harvesting and thawed up again for lysis, and 1 mM PMSF was added to the lysis suspension. After sucrose gradient purification fractions containing protein were stored at 4 °C. Two years later, the sample was concentrated in a 100 kDa cutoff centrifugal filter in TRIS buffer (50 mM TRIS, 50 mM NaCl, pH adjusted to 7.5). A fraction of the protein was diluted in acetate pH 5.5 buffer (50 mM sodium acetate*3H2O + acetic acid until pH 5.5, 50 mM NaCl) and concentrated again in a 100 kDa cutoff centrifugal filter until the buffer was exchanged in a stepwise manner to acetate pH 5.5 buffer. The pH 5.5 sample was split into two parts and the buffer one of the pH 5.5 samples was then again exchanged for pH 7.5 TRIS buffer (50 mM TRIS, 50 mM NaCl, pH adjusted to 7.5).

## Solid-state magic-angle spinning NMR spectroscopy and resonance assignment

Carbon-detected solid-state NMR spectra of $^{13}C$–$^{15}N$ uniformly labeled ACNDV capsids were recorded on an Avance III Bruker wide-bore 850 MHz spectrometer. Protein samples were spun in NMR rotors of 3.2 mm diameter at 17 kHz MAS frequency. The sample temperature was 3–5 °C[63] and chemical shifts were referenced to sodium trimethylsilylpropanesulfonate (DSS). The protein resonances were assigned using carbon-detected 2D and 3D correlation spectra: $^{13}C$–$^{13}C$ DARR, NCA, NCO, NCACB, NCACX, NCOCX, CANCO, CCC, CANcoCA, NcoCACB (Tables S1, S2, and S3). The assignment strategy as described in references[64] and[65] was employed. Spectra were processed with Bruker TOPSPIN software version 3.5 and analyzed with CcpNmr Version 2[66]. The assignment is deposited in the BMRB databank under accession number 51506.

Proton-detected spectra of fully protonated $^{13}C$-$^{15}N$ labeled ACNDV capsid protein at pH 7.5 were recorded in NMR rotors with 0.7 mm diameter at 100 kHz MAS on an Avance III Bruker wide-bore 850 MHz spectrometer. The sample temperature was determined to 19–22 °C[63] and chemical shifts were referenced to DSS. For the assignment of NH amide and HA proton resonances, 2D spectra (hNH, hCH) and 3D spectra (hNCAH, hCANH, hCAcoNH, hNcoCAH) were recorded (Tables S1, S2, and S3)[67]. The already known resonances from carbon-detection experiments were used for verification of the assignment and for identification of proton resonances where the connection in the sequence was difficult to assess.

## Transmission electron microscopy

Formvar-copper 400 mesh transmission electron microscopy (TEM) grids were treated by glow discharge for 1 min at 25 mA, negative

polarity. A grid was laid onto a drop of 5–10 µL sample solution and blotted after 1 min. The grids were placed on a drop of 2% uranyl acetate and stained for 15 s. TEM images were obtained on a Hitachi HT7700 EXALENS instrument at 100 keV with a tungsten/LaB$_6$ emitter. The microscope was equipped with an 8 M pixel CCD camera.

## Grid preparation for cryo-EM

Protein samples were expressed as described above. For measurements at pH 7.5, cryo-EM grids (Quantifoil R2/2 Cu300) were incubated overnight on filter paper soaked with ethyl acetate subsequently covered with 1 nm continuous carbon film by floatation. The grids were glow discharged for 15 s at 15 mA. Grids were vitrified in an ethane/propane mixture with a Vitrobot plunge freezer at 4 °C and 100% humidity. 3.5 µL of sample solution was applied to the grid and the wait time was set to 60 s. Blotting times ranged from 6 to 16 s. The preparation of the pH 5.5 sample was carried out as for the pH 7.5 sample with following differences: Glow discharge was performed for 30 s with 25 mA and 4 µL of sample were deposited on the grid for vitrification.

## Cryo-EM data collection and image processing

Vitrified grids of the capsid at pH 7.5 were imaged with a Titan Krios electron microscope (Thermo Scientific) operated at 300 kV using a K2 direct electron detector (Gatan) in counting mode behind an energy filter. The energy filter slit width was set to 20 eV. 8999 movies were recorded with the EPU data acquisition software at a defocus ranging between −0.5 µm and −2 µm at 130'000x magnification (pixel size 1.07 Å/pixel) with an electron dose of 76 e − /Å$^2$ fractionated over 40 frames. After drift correction and dose weighting with MotionCorr[68], particles were picked with crYOLO 1.7.6[69] and defocus was determined with Gctf[70]. Micrographs of the capsid at pH 7.5 were manually selected based on the quality of the power spectrum and micrographs recorded with defocus values smaller than −0.5 µm were removed. 375,654 particles were selected with crYOLO. Further image processing was carried out in Relion 3.1[71]. Particles were extracted without binning (box size: 432 pixels, pixel size: 1.084 Å/pixel) and subjected to 3D classification into 10 classes without application of symmetry, using an initial model generated in Relion. One class (138,158 particles) displayed the features expected of a viral capsid with icosahedral symmetry. This class was subjected to a second round of 3D classification into 10 classes with application of icosahedral symmetry, with 81,855 particles in the selected class. The second 3D classification was performed over 100 iterations and particles that changed class more than 40 times were excluded from the dataset, resulting in 70,868 particles. Refinement with application of icosahedral symmetry resulted in a map with an overall resolution of 3.7 Å (Fig. S6). For symmetry expansion of the capsid at pH 7.5 we followed an overall similar approach, but initially included micrographs with lower defocus, performed two additional rounds of 2D classification, used initial model generation (5 classes, without application of symmetry), and subsequent 3D classification into 10 classes without application of symmetry (5 classes selected, 111,808 particles). Refinement with application of icosahedral symmetry yielded a map with a resolution of 3.7 Å. A mask around the asymmetric unit was generated with Phenix[68] and the refined particles were symmetry expanded in Relion, yielding 6,708,480 asymmetric unit particles. The symmetry expanded asymmetric unit particles were re-extracted with a box size of 128 pixels without binning, recentered, and classified into 25 classes without alignment. Class 16 contained 1,895,898 particles and refined to a map with an overall resolution of 3.8 Å. Despite of nominally lower resolution, the map of the asymmetric unit showed superior detail and was used for model building.

Vitrified grids of the capsid at pH 5.5 were imaged with a Titan Krios electron microscope using a Falcon 3 direct electron detector (Thermo Scientific) in integrating mode. 5362 movies were recorded with a defocus ranging between −0.2 µm and −2 µm at 165,000× magnification (pixel size 0.84 Å/pixel) with an electron dose of 80 e − /Å$^2$ fractionated over 40 frames. After drift correction and dose weighting with MotionCorr[64], particles were picked with crYOLO 1.7.6[65] and defocus was determined with Gctf[66]. 215,225 particles of the capsid were extracted with a box size of 512 pixels, scaled to 64 pixels (pixel size: 6.6 Å/pixel), and subjected to one round of 2D classification in order to remove junk classes as well as overlapping and deformed particles, leaving 163,119 particles. Particles were extracted with a box size of 432 pixels without binning and re-centered. The map of ACNDV capsid at pH 7.5 was used as initial model for 3D classification into 10 classes without application of symmetry. A refined map arising from class 8 (85,303 particles) showed the expected features. The particles in this class were subjected to a 3D classification into ten classes with application of icosahedral symmetry. The most populated class (class 10, 83,424 particles) refined to a map with a resolution of 4.2 Å. Analysis of the class distribution during classification revealed that a subset of particles oscillated between class 10 and class 1. Removal of particles that change class more than six times during 100 iterations of 3D classification from class 10 retained 77,129 particles and improved the resolution of the refined map to 3.9 Å (Fig. S8).Cryo-EM measurement settings are summarized in Table S8.

**Cryo-EM experiments for particle size determination.** Samples were vitrified on Quantifoil R2/2 grids coated with 1 nm continuous carbon, using a Thermo Scientific Vitrobot Mark IV. The climate chamber was set to 4 °C at 95% humidity. Grids were incubated for 30 s pre-blot and blotted once for 2 s or 3 s. Grids were imaged with a Thermo Scientific Titan Krios using a Falcon 3 direct electron detector in linear mode. Micrographs were acquired at a magnification of 129,000× (resulting calibrated pixel size 1.087 Å/pixel) with an exposure time of 0.77 s and fractionated over 30 frames. A total number of 1515 micrographs were acquired for sample 1 (pH 7.5), 4023 for sample 2 (pH 5.5), and 2384 for sample 3 (pH shift from pH 5.5 back to pH 7.5).

Image processing was carried out in CryoSPARC[72]. Micrograph movies were drift corrected and averaged using cryoSPARC patch motion correction with default settings. Defocus was determined using cryoSPARC Patch CTF estimation with default settings. Capsids were selected manually from 10 micrographs of sample 1 and the resulting particle positions were used to train a neural network with Topaz[73] in order to pick particles with high accuracy. All capsids were selected independently of the shape and size during manual picking to achieve a representative image on the particle population. The same neural network model was used for picking particles from all three datasets. Picking identified 146,077 particles in dataset 1, 346,287 particles in dataset 2, and 257,179 particles in dataset 3. Particles were extracted with binning (box size 512 pixels to 96 pixels) and subjected to 2D classification into 50 classes in cryoSPARC. After a first round of classification, classes that were empty or showed only fragments of a capsid molecule were removed. The relatively low number of around 10% of all particles removed during the first round of 2D classifications (Sample 1: 10,709 particles rejected, 135,368 particles accepted; sample 2: 38,274 particles rejected, 308,013 particles accepted; sample 3: 24,036 particles rejected, 233,143 particles accepted) attests to the high quality of neural network picking. The remaining classes were subjected to a second 2D classification into 50 classes The particle acceptance rate for the samples 1 and 3 (both at pH7.5) were very similar with 29% and 33%, respectively, while the acceptance rate for the sample at pH5.5 was substantially higher at 50% (Sample 1: 95,999 particles rejected, 39,369 particles accepted; sample 2: 155,356 particles rejected, 152,657 particles accepted; sample 3: 157,284 particles rejected, 75,859 particles accepted). Classes that contained particles that resemble the expected assembled Nackednavirus capsid with an intact shell were selected and subjected to a third round of 2D classification.

Selected particles were re-extracted with a box size of 512 pixels scaled to 256 pixels (resulting pixel size: 2.17 Å/pixel). Initial 3D models from all samples were generated in cryoSPARC using ab-initio model generation with one class and applying icosahedral symmetry. A set of 100 (sample 1) and 1000 (sample 3) particles was sufficient for a high-quality initial model that was subsequently used as starting model for homogeneous 3D refinement in cryoSPARC, also with application of icosahedral symmetry. For sample 2, initial model generation was only successful when 1000 particles and three classes were used. Homogeneous refinement resulted in maps with a resolution of 4.53 Å (sample 1), 4.15 Å (sample 2), and 4.10 Å (sample 3). Local refinement with tight masks that remove the unstructured nucleic acids close to the inner surface of the capsid shell further improved the resolution to 4.1 Å (pH 7.5), 3.8 Å (pH 5.5), and 3.9 Å (pH shift from pH 5.5 to pH 7.5). Maps were aligned in Chimera[74].

### Protein model building and structure refinement

An initial model of the ACNDV capsid protein monomer was built with *Coot* version 0.8.9[75] where helices were placed into the cryo-EM density map obtained at pH 7.5 (Fig. 2b). The positions of tryptophan, tyrosine, and phenylalanine residues were identified by the apparent sidechain density. The secondary structure elements from the analysis of solid-state NMR chemical shifts were implemented in the initial model. The asymmetric unit was built from three monomers. In *Phenix* version 1.20.1-4487[76] the asymmetric unit was refined with the "real space refine"[77] routine. Hereby, the refinement strategy included global minimization, rigid body fitting, and atomic displacement parameters refinement. The asymmetric unit was expanded to the whole particle with PyMOL[78]. The asymmetric unit was further refined with ISOLDE 1.0b4.dev0[79], an interactive structure determination tool available as UCSF ChimeraX plug-in[80]. Bad rotamers and bad cis/trans-conformations were manually corrected in ISOLDE applying the default settings (AMBER 14 force field). For the model at pH 7.5, the cryo-EM density map of the asymmetric unit obtained by symmetry expansion was used to improve sidechain orientations. In the map of the asymmetric unit (resolution of 3.8 Å) sidechain density is better visible than in the full capsid map.

The protein model of the ACNDV capsid at pH 5.5 was built and refined in the same way as the pH 7.5 structural model. The asymmetric unit of pH 7.5 capsids was used as starting point. Iterative refinement was again performed with ISOLDE and *Phenix*. For the refinement procedure a cryo-EM density map of the entire capsid (Fig. 2e, 3.9 Å resolution, FSC cutoff at 0.143) was used.

Validation parameters from MolProbity[81–83] for the final models at pH 7.5 and at pH 5.5 are given in Table S4 and details of the fitted models are shown in Fig. S9. N-terminal and spike residues, as well as hydrogen atoms were only removed after structure refinement. The structures were submitted to the Protein Data Bank. In this publication, protein structures are visualized with UCSF ChimeraX[80].

### Reporting summary

Further information on research design is available in the Nature Portfolio Reporting Summary linked to this article.

## Data availability

Chemical shifts of $^{13}C$, $^{15}N$, and $^1H$ atoms of ACNDV Cp at pH 7.5 were deposited in the Biological Magnetic Resonance Data Bank under accession number 51506. Cryo-EM maps of ACNDV capsids at pH 7.5 and at pH 5.5 were deposited in the Electron Microscopy Data Bank under accession numbers EMD-15295 (pH 7.5) and EMD-16371 (pH 5.5) and molecular capsid structures were submitted to the Protein Data Bank under accession numbers 8AAC (pH 7.5) and 8C0O, (pH 5.5). Two- and 3-dimensional NMR spectra are available upon request. Source data are provided with this paper.

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

## Acknowledgements

We thank Riccardo Cadalbert and Simon Widler for their effort in teaching cell culture handling and for their great support with protein expression. Simona Rodighiero, Sebastian Tacke, and Cecilia Bebeacua, former or present members of ScopeM at ETH Zurich, were involved in measurements with the transmission electron microscopes, for which we are very grateful. Further, we would like to thank Pavel Afanasyev, member of the Cryo-EM Knowledge hub, at ETH Zurich and Miroslav Peterek of ScopeM at ETH Zurich for their support in cryo-EM data analysis. We also thank Marc André Leibundgut from the Institute of Molecular Biology and Biophysics at ETH Zurich for his help in structure refinement. This research was supported by an ERC Advanced Grant (grant number 741863, Faster to B.H.M.) and by the Swiss National Science Foundation (grant number 200020_159707 and 200020_188711 to B.H.M.) and by the French Agence Nationale de Recherche (ANR-14-CE09-0024B) and the LABEX ECOFECT (ANR-11-LABX-0048) within the Université de Lyon program Investissements d'Avenir (ANR-11-IDEX-0007) to A.B.

## Author contributions

A.B., M.N., D.B., and B.H.M. conceived the study and, together with S.P., designed the experiments. S.P. with the support of T.W. and A.A.M. recorded NMR spectra. S.P. prepared the samples with support of L.L. and M.N. S.M. screened the initial samples by cryo-EM. J.R. recorded cyro-EM data. J.R., D.B., and S.P. processed the EM data. S.P., J.R., A.B., D.B., and B.H.M. analysed data and wrote the manuscript with S.P. providing the first draft. S.P., S.S., R.B., A.B., M.N., D.B., and B.H.M. interpreted data. All authors read and approved the final manuscript.

## Competing interests

The authors declare no competing interests.

## Additional information

[1]Physical Chemistry, ETH Zurich, 8093 Zurich, Switzerland. [2]Cryo-EM Knowledge hub, ETH Zurich, 8093 Zurich, Switzerland. [3]EMBL Imaging Centre, European Molecular Biology Laboratory, EMBL Heidelberg, 69117 Heidelberg, Germany. [4]Molecular Microbiology and Structural Biochemistry, UMR 5086 CNRS, Université de Lyon, 69367 Lyon, France. [5]Division of Virus-Associated Carcinogenesis (F170), German Cancer Research Center (DKFZ), 69120 Heidelberg, Germany. [6]Department of Infectious Diseases, Molecular Virology, University of Heidelberg, 69120 Heidelberg, Germany. [7]Department of Medicine II / Molecular Biology, University of Freiburg, Freiburg im Breisgau, Germany. [8]Present address: Max-Planck-Institute for Chemical Energy Conversion, Stiftstr. 34-36, 45470 Mülheim an der Ruhr, Germany. [9]Present address: Institute of Technical and Macromolecular Chemistry, RWTH Aachen University, Worringerweg 2, 52074 Aachen, Germany. ✉e-mail: a.bockmann@ibcp.cfr; michael.nassal@uniklinik-freiburg.de; boehringer@mol.biol.ethz.ch; beme@ethz.ch

