## [Peer Review File · Nature Communications]

Reviewers' Comments:

Reviewer #1:

Remarks to the Author:

African cichlid nakednavirus (ACNDV) is a none-enveloped virus that is evolutionary related to the enveloped Hepadnaviridae. Pfister et al. present structural insights into the organization of virus-like particles of ACNDV using electron microscopy and NMR. They obtain maps with some 3.7-3.9 Å resolution, which allow them to model large parts of the assembly domain of the capsid protein. Their structural findings confirm the previous structural model by Lauber et al. 2016 at 8 Å resolution, in which the arrangement of the subunits in a T=3 capsid was revealed and the structural similarity to the human Hepatitis B core Antigen (HBc) was established. However, the better resolved maps allow Pfister et al. to derive structural models, which show distinct differences in the stabilization of the inner and inter dimer packing of the subunits in the capsid compared to human HBc. One interesting aspect of their work is that they observe a significant shrinkage of the capsids by lowering the pH to 5.5 which they speculate might be "reminiscent of structural changes upon viral cell entry".

The structural work broadens our understanding of the organization of ACNDV capsids and is a valuable add-on to the existing knowledge. One strength of the manuscript is the detailed description of the methods and materials, which document the differences in the handling of the sample for NMR and for cryo-EM and at pH 5.5 and pH 7.5.

The introduction reads very exciting and appealing to a wide audience and phrases the fundamental question for the structural adaptation to a non-enveloped life

style in nakednaviruses. However, these expectations are not met by the structural findings in the results part. The discussion and final conclusions address the questions from the introduction again but the discussion remains largely speculative and not well supported by the experimental findings. Similarly, the conclusion about the lower charge in the interior enabling uncoating earlier in the viral life cycle is not supported by any experimental evidence. It is not even known whether an earlier uncoating takes place in ACNDV. This disconnects the solid results from the highly speculative elements of the discussion and conclusion.

Similarly disconnected is the use of NMR and electron cryo-microscopy for the structural investigations. It is not always clear whether the NMR-studies and the electron cryo microscopy give the same findings. Whether they are complementary or whether they even contradict each other. For example, NMR shows aggregation over time and at pH 5.5 and electron microscopy shows that there is no aggregation. Assigned start- and end-points of the helices differ in NMR and in electron microscopy. While these discrepancies are not a problem per se, they add concerns about the statements drawn on packing of side chains, structural dynamics or pH dependent stability and conformational changes.

More specific comments

Introduction

1. Line 65-72 "likely to employ fundamentally different infection and virion release mechanisms" It was shown that naked human HBc particles are secreted in an Alix dependent manner (Bardens et al. 2011 Cell Microbiol). Therefore, it is conceivable that an envelopment independent capsid release mechanism is encoded in the capsid structure of human HBc. It is similarly likely that such a release mechanism is structurally preserved. Other capsid specific mechanisms of the viral life cycle such as the need for genome-release, the transport of capsids within the cell or the escape from the endosome are very similar for enveloped and none-enveloped Hepadna- and Nakedna viruses. Taken together, it is probably less likely that the mechanisms in nakedna viruses are so fundamentally different as suggested.

Results:

2. Lines 143-145 "Information about helix positions and lengths allowed for a fast de novo building of an initial structure"

Start positions and the end positions of the helices and the length of the helices derived from NMR and cryo-EM are somewhat different. Out of curiosity: is this information from NMR so much more beneficial for denovo model building than the existing predictions from homology modelling and the secondary structure predictions?

3. Lines 153-155 "Apart from signs for protein aggregation,,three additional peaks were observed in the pH 5.5 spectrum, including resonances of histidine and isoleucine residues (Figure S 3D)" Is it clear where these three differences are located in the structure. Is their evidence from the

cryo-EM data, which indicates a structural change close to these residues. Does cryo-EM see any effect of His-protonation on the structure?

4. Line 182: "the very first three residues are not visible in the cryo-EM density map"

Is the C-term not resolved because of the additional Gly? If so, is there any indication that the α -helix continues to the very N-terminus?

5. Lines 189-190: "Comparison of the AB dimer, formed by the quasi-equivalent conformation of chains A and B, with the CC dimer, reveals differences in the dimer interface formed by the spike helices (Figure 3E,F compared to Figure S 8F,G)"

It is not clear, which differences are meant. The figure is very busy and it is difficult to see the interactions in the figure. It is unclear, which differences in the interaction network are real and which are related to modelling errors. At a resolution of 3.7-3.9 Å side chain density is not very reliable. No data is shown that shows whether the difference in the models is supported by the density in the map. Many differences map to smaller side chains. At the given resolution their rotamer orientations cannot be assigned without ambiguity. Other differences map to negatively charged residues such as Glutamate. These residues are typically not resolved in the EM-density. Therefore, the side chain assignment of these residues is very ambiguous. In addition, the upper spike is less well resolved and does not show any continuous density. Therefore, differences in the interaction network in the upper spike region are unlikely to be mapped with confidence. To support their claim the authors should show that the differences in the model are fully supported by the map density and that there is evidence from the NMR data, which should map a different environment for these residues.

6. Lines 194-195: " Thus, the N-terminus in the CC dimer likely stabilizes a conformation in which the core fold of the capsid protein (hand region and spike helices) is rotated relative to the dimer interface"

It is unclear what is meant by the "dimer interface"? Is this the 2-fold symmetry axis in the CC-dimer (intra dimer) or the interface between dimers (inter)? Please annotate the relative rotations in the figure.

7. Line 198: "The N-terminus with its α -helix contacts adjacent helices of the partner molecule in the dimer"

How much of the inner dimer interface is contributed by the N-terminal α -helix compared to the core helices in the spike?

8. Lines 206-207: " We observed that the capsid diameter is smaller at pH 5.5 (21.7 nm, Figure 2 H) than at pH 7.5 (23.0 nm, Figure 2 D) yet, the asymmetric units at pH 7.5 and at pH 5.5 appear very similar."

Variable diameters of human Hbc are frequently observed in 3D-classification regardless of an external cue. In this analysis of capsid-like particles at pH 7.5 and 5.5 a very small fraction of the cryo selected particles was retained in the final map (according to Figure S2 10% at pH 7.5 and 36% at pH 5.5... these numbers do not agree with the numbers quoted in the Materials and Methods. Please check!). Therefore, the question remains whether the change in diameter is related to a change in pH or to a selection of different subsets of the particle population or some other coincidence.

Another complication along these lines is that the data was collected with two very different camera set-ups (K2 vs Falcon III). These cameras have different pixel sizes and are also located at different positions in the microscope. The pixel size depends very much on the calibration and can easily have an error in the order of 5%. As this is the difference, which is observed between the capsids at pH 5.5 and pH 7.5, it is necessary to explain how the two microscope set-ups were calibrated, whether they have been calibrated with the same standard (e.g. against map of ferritin) and what the likely error in the calibration procedure was.

Measuring the diameter in the maps can be quite cumbersome. Therefore, the authors chose two distinct atoms in the model of the subunits in the capsid. This gives a very accurate measure of the diameter if the model is accurate. However, in table S4 Mtriage quotes a map model CC of 0.5 at 7.4 Å for the model at pH 5.5 (please note the threshold of 0.5 is decisive threshold when comparing map and model). This suggests that either the map has no resolution of $3 \times$ Å or that the model is wrong. Please check! and improve the map to model correlation.

To claim that there is a pH dependent change in diameter the authors should 1) calibrate the microscopes to a common standard (or collect the data with the same set-up); 2) show that the change in diameter reverses when increasing the pH again. As such large changes in diameter can be mapped at some 6-8 Å resolution, it is advised to establish the changes with small scouting experiment where one preparation of ACNDV is buffer exchanged on the concentrator to pH 7.5

then exchanged to 5,5 and then back to 7.5. For all three conditions cryo-EM data should be acquired at the same microscope.

9. Lines 208-209: "Chain B can be superimposed almost perfectly at either pH, while chains A and C are slightly shifted relative to chain B."

Is "perfectly" because chain B was superposed with Matchmaker? Would the conclusion differ if chain C would be superposed? Please clarify (e.g. in the figure legend for 2k)!

10. Lines 230-231: "RMSD of CA atoms comparing superimposed AB and CC dimers with largest RMSD values at the hand regions."

The RMSD seems to be very much dependent on how the superposition was done. If the match would be optimized for the hand region the rmsd values would probably be much smaller and in the Spike region within the error of modelling (1-2 Å). Similarly, if the AB- and CC-dimers could be fitted into a common molmap of a dimer before calculating the RMSD values, they are probably also much lower. This different way of fitting to a common map would generate a more equal distribution of errors and would probably hint at a less stunning difference of the hand-region.

11. Lines 272-273: "Unassigned cryo-EM map density located below the spikes of AB dimers but not below CC dimers was found in the ACNDV capsid maps (Figure S 11). Likely sources of the diffuse density below AB dimers are enclosed RNA (solid-state NMR identifies nucleic acids to be RNA and not DNA [data not shown]) and/or the flexible CTD of the capsid protein"

The cryo-EM density is shown in Figure S12! and appears very speckled. Therefore, it is unclear what the relevance of this density pattern is. This should probably be low-pass filtered for better visibility.

Considering that charges, RNA and DNA become a major part of the discussion, the following points should be clarified: "1) How much RNA is incorporated in the capsid shells per protein. It should be possible to estimate the ratio from Agarose gels stained for RNA and protein.

Alternatively, the ratio can also be compared spectroscopically. Is the ratio different to human Hbc? 2) Does the EM-map give a radial density profile, which shows that incorporated density aggregates in distinct shells?. If so, what is the distance between the peak of the protein shell and the "RNA-shell". For human Hbc the distance is very characteristic and there is a clear gap between protein shell and "RNA-shell". This reviewer expects that In ACNDV the distance is smaller because many of the positive charges are accumulated on the inner capsid surface (see Figure S13) underneath the spikes rather than in the unresolved CTD.

There could also be other mechanisms of packaging and charge compensation which might generate a more diffuse density distribution in the capsid interior.

12. Lines 287-288: "a quicker aggregation within the NMR rotor suggests that pH 5.5 capsids are less stable than pH 7.5 capsids."

This statement confuses solubility with stability. The authors observe solubility. They show in their supplemental material, that negatively stained capsids in "aggregated" samples look intact without indication for denaturation. They are probably even able to dissolve the aggregates again as they do not see aggregation at the lower concentrations required for negative stain.

The authors do not show a reduced stability of the capsids. To show a change in stability they should do a Thermo-shift-assay, which shows a different Temperature for the onset of denaturation. To relate stability to a folding enthalpy, more sophisticated experiments such as differential scanning calorimetry would be necessary.

This reviewer does not understand how an observation of solubility relates to possible conformational changes or decapsidation in the subsequent discussion.

13. Lines 292-300: "The evolutionary lineages..."

In the context of evolutionary lineages the authors should interrogate the structure for the inter-dimer contacts, which have three conserved motives according to Dill et al. J. Vir 2016. Whereas the other landmark features such as inner dimer contact, tips of the spike, C-term, N-term are not conserved between Hepadnaviridae including Nakednaviridae,

14. Lines 316-319: "Plausibly, the reduced number of arginine sidechains may be unable to electrostatically compensate for a largely double-stranded full-length DNA such that the capsid shells are disrupted early during second strand DNA synthesis; conversely, particles containing shorter reverse transcription products, such as from the major 2.0 kb SP1 splice product, would still be stable."

This discussion of the second strand formation is purely speculative. The mechanism of charge balance in ACNDV is unknown. Charges could also be compensated by incorporating divalent cations such as Magnesium or small molecules like spermidine. Human Hbc inhibits the interaction of the C-term with unspecific RNA by phosphorylation. However, there are much less serins in the

ARD of ACNDV (3) compared to human HBC (7), which could be potentially phosphorylated. Therefore, the charge balance mechanism might vary rather than modulating the second DNA strand formation or disrupting the capsid.

It is also unclear what the importance of disrupting a capsid during formation of the 2nd DNA strand would be. This would basically interfere with the newly formed virus before it egresses from the infected cell. However, decapsidation would make more sense for a mature capsid after it entered the cell and has targeted the nucleus.

Without any experimental evidence for a formation of a shorter second DNA-strand in ACNDV, this discussion is too speculative to be useful – rephrase and support by evidence (either experimental or via references)

15. Lines 329-333: "Still, the packaging capacity would be smaller for ACNDV Cp because it exclusively assembles into T=3 capsids which means that 25 % less protein is available for neutralization of nucleic acid phosphates. Since genomes of HBV and nakednaviruses are of comparable size of around 3'000 bp, the genomes of nakednaviruses appear too long to form stable nucleocapsids and thus, reverse transcription might cease at an earlier stage in ACNDV nucleocapsids as compared to HBV."

This is an interesting aspect and very much along lines 316-319. However, it is not known whether ACNDV also forms exclusively T=3 capsids in mature viruses. It could well be that the viral polymerase together with the RNA-genome nucleate the assembly of larger capsids such as the T=4 capsids. Again, 2nd strand synthesis could be tested in cellular assays.

16. Lines 340-341: "Also, the smaller opening of the ACNDV capsid triangular pores (in the center of Figure 6A) are compensated by bigger openings at the threefold axes"

Whether something can pass holes very much depends on the properties of the holes in addition to the size. What are the properties of the holes at the three-fold axes (e.g. hydrophobic, hydrophilic, charged?), Are these properties any different in human HBC? Figure 6A is one of the few places in the manuscript, where the map density is shown. The threshold is chosen that the map looks very skinny and the holes might appear exaggerated. The holes look somewhat larger than in the space fill representation of the model in Figure 2. It would make sense to check the properties of the holes at the local and strict 3-fold symmetry axes for example with the program "HOLE" to get further insights into the significance of the difference in apparent hole size.

Lines 474 -478: "Cp-like large and accessible hydrophobic area within the capsid spikes in ACNDV Cp implies that the hydrophobic pocket in HBV Cp evolved after the separation of hepadna- and nakednaviruses. While still hypothetical, binding of a natural pocket factor into the HBV Cp hydrophobic pocket may be related to the capsid maturation and envelopment [48, 55,56]. Hence, the absence of an equivalent pocket in ACNDV Cp might well reflect that the HBV Cp hydrophobic pocket co-evolved with the acquisition of envelopment.

The authors compare structural elements of ACNDV mainly with human HBC. They find that many important structural determinants are not conserved, such as the stabilization of the dimer interface, the disulphide bond and an enigmatic hydrophobic pocket in. However, the side-by-side display of the monomers of DHBC, HBC and ACNDV in Figure 5 suggests that the packing of the helices in the core spike without the extension is much more similar to DHBC than to HBC. In particular helix 3 is tilted away from the dimer axis in DHBC and in ACNDV but not in HBC and helix 4 is straight in ACNDV and DHBC but kinked in HBC. Like ACNDV, DHBC-dimers are not stabilized by an SS-bridge, their inner dimer interface is more hydrophilic, they have a α^+ helix, although in a different orientation and they do not have a hydrophobic pocket. It would be helpful to extend the comparison in a similar way towards DHBC (e.g. add a comparable panel for DHBC to figure S14), In the light of a missing hydrophobic pocket in DHBC, this feature seems to be unrelated to envelopment in general.

The analysis of Lauber et al. does not suggest that ACNDV is more similar to HBC than to DHBC.

Conclusions:

17. Lines 484-485 Conclusion: " although envelopment implies fundamental differences in virus life-style, including host and tissue tropism, its acquisition had only a minor effect on the capsid protein fold.

Most of the differences in host and cell tropism are related to the envelope and not to the structure of the capsid. Egress of non enveloped capsids is maintained in HBC (see above). Therefore, the differences in virus-life style in respect to the capsid are probably not so fundamental as suggested by the authors. Thus, it is unsurprising that the general fold of the capsid is maintained.

18. Lines 487-490: "Given the similar genome sizes of about 3 kb of both nakedna- and hepadnaviruses this implies that formation of the double-stranded DNA genome inside the capsids

exerts more internal pressure on ACNDV than HBV capsids, hence genome uncoating may take place earlier in the viral life-cycle of nakednaviruses.”

I do not understand how this can be concluded from the structural work considering that 1) It is unknown whether T=3 capsids are the predominant form for mature virus particles; 2) whether DNA strand formation leads to decapsidation. What is meant by earlier in the viral life-cycle? Does this mean that decapsidation takes place in the cytosol before the release of the mature virus from the cell? This does not seem to be a plausible model that is in any way supported from the structural findings.

19. Lines 490-491: “the assembled icosahedral viral particles were found to be, in vitro, less resistant against denaturation than HBV, as seen by NMR.”

The authors show that the capsids are less soluble but not less resistant against denaturation. In addition, the negatively stained controls at a much lower concentration suggest that the aggregation is reversible by dilution, as they do not observe it. This again points towards solubility rather than denaturation.

Materials and Methods:

20. Lines 573-576

Repetition of the start of the previous paragraph (lines 555-558).

Validation Reports

The validation reports are incomplete (no clash report). The reports do not show the map to model validation. The validation reports shows that residues 56-75 are missing, whereas only 66-75 were reported missing in the results of the paper > The validation does not match the represented results.

Supplemental Material

21. Table S4

The table indicates a low similarity between map and model for the pH5.5 structure (CC 0.5 map vs model =7.4A). This is unexpectedly bad. Show a detailed presentation, which supports the quality of the model and its agreement with the map.

It would be nice to either extend the table or have another table for the details on the imaging and image analysis (microscope, detector, magnification pixel size, focus range,,total exposure, exposure rate, linear vs counted, number of selected particle, number of particles in final map, resolution according to FSC...)

Out of curiosity: Did the Krios with K2 had no energy filter? If not add information on the energy filter settings.

22. Figure S2

The figure is too small to recognize the details. > Make two separate figures for pH 5.5 and 7.5.

The numbers of particles do not match the numbers in the materials and methods in the main paper... check and correct

FSC-curves are missing for documenting the quality of the maps. Gallery of details of map with fitted model are missing for documenting the quality of the map and the fidelity of the model.

23. A8 D-F

This is very busy and hard to see. Try thinner slabs.

24. S10

Try local resolution filtering of the map (or deep learning density modifications) for a more adequate representation of the upper spike region.

25. Figure S12

This figure shows that many side chains in the model are not accommodated by the EM-density. This raises concerns about the validity of the side chain orientations in the model and subsequently the interaction/contact networks shown in Figure 3 and S8.. At a resolution of only somewhat better than 4A, a discussion of changes in such interaction networks seems to be overstating the structural information.

Reviewer #2:

Remarks to the Author:

The manuscript by Pfister et al builds on previous work by co-authors Stefan Seitz and Ralf Bartenschlager on the evolution of the hepatitis B virus (HBV) and its relation to naked viruses with a very similar capsid and the same genome replication strategy. The previous work had reported a cryoEM structure at 8Å resolution of the African cichlid nakednavirus (ACNDV) using

recombinant capsid protein produced in *E. Coli*, which spontaneously assembles into icosahedral capsids with triangulation number $T=3$. Here, the authors used a combination of single-particle cryo-EM to a resolution of 3.7\AA and magic-angle-spinning solid-state nuclear magnetic resonance (MAS ssNMR) to provide a full atomic model of the ACNDV capsid. The authors attempt to identify potential clues to understand how the naked capsid of ACNDV might deliver the viral genome into the target cells nucleocapsid, a process that in the case of HBV takes place via membrane fusion, which in principle is not possible for a naked virus particle. This leads them to determine the structure of the recombinant ACNDV capsid at pH 5.5 as well as pH 7.5 (accompanied by determination of the MAS-ssNMR structure), which shows that the particles shrink at acid pH to a diameter of 217\AA instead of 230\AA (close to 6% shrinkage, which means an 11% reduction in viral surface area, and 16% shrinking of the total enclosed volume. Yet they do not observe any particular role of pH in particle destabilization, which could be a way of disassembly upon uptake into an endosome. They observed however a tendency of aggregation of the particles at low pH, which they can also see in the MAS-ssNMR experiments. The authors also examine the particles at pH 9 by MAS-ssNMR, without observing any major differences.

Overall, the results reported are important as they provide an atomic model for the Nakednaviruses that was not available. I have, however, issues with the way the manuscript is presented, as the authors do not seem to use the resulting model correctly in order to try to identify all potential clues for virus entry they claim to be searching. Also, the text appears at moments too technical, as if the paper was destined to specialists in the techniques being used (in particular, NMR), rather than to the average reader of *Nature Communications*. In particular, the fact that NMR is capable of providing information on the dynamics of a particle, which are absent in static structures such as those obtained by X-ray crystallography or by cryo-EM, is not presented at all.

For instance, the reader is shown ^{13}C - ^{13}C DARR correlation spectra or hNH spectra, without any attempt to introduce them to the reader. The paper would greatly benefit from explaining the rationale for the dual approach taken for the structural studies, that the resolution of 3.7\AA is limiting in the side chain assignment to build a full model, and that the complementarity introduced by NMR is very important to obtain a reliable atomic model in this case. Also, that standard solution NMR is not applicable to study particles of the size of the ACNDV capsids, but that the powerful methods of solid-state NMR measure while spinning the sample at high speed at the magic angle makes it feasible. They should also explain that like in solution NMR, it is necessary to use certain isotopes (in particular ^{13}C), and that DARR stands for Dipolar Assisted Rotational Resonance, in which the magnetization is transferred from hydrogen (^1H) to ^{13}C nuclei, from where it is further transferred to other ^{13}C nuclei which are close in space, in a process that allows to build an atomic model from the resulting spectra. And that hNH spectra inform on the magnetization transferred from ^1H to ^{15}N via cross-polarization, which is also important to build the model. It is not necessary to give a full course on NMR, but just enough explanation so that the reader understands the meaning of the spectra that are shown. In addition, they should highlight the impact of information given by split peaks for instance, where there are different interactions in the various environment of a given residue within the capsid. As it is, the authors mention that the tryptophan residues of the ACNDV capsid give rise to split peaks but they do not explain what information it is providing about the dynamics of the structure, or about alternative conformation on the three different environments of the T-3 capsid,

In addition to the issues related to the presentation of their results, the authors do not seem to exploit the atomic model to get extensive information. For instance, does any hydrophobic patch appear on the capsid in the pH 5.5 form that is absent in the pH 7.5 form, which could explain the observed aggregation? They do show the surface hydrophobicity distribution for an AB dimer in Fig. S14, but it is on the whole particle surface that this should be shown, if they are to identify a potentially functional patch. Similarly, can they calculate the surface electrostatic potential of the particles at the two pH values, to see if there is a different distribution in the two? Again, they show the surface electrostatic potential for the inner surface of the AB dimer in Fig. S13, but it is the external surface of the whole particle that would be relevant to look at. In addition, if there are by now sufficient sequences from distant nakednaviruses, the authors could also try to find a conserved surface patch by using the multiple sequence alignments, which could point to a potential site for receptor binding.

Another issue is about their interpretation of the protection of nakednaviruses against environmental degradation of the genome, as normally the capsids of naked viruses are totally sealed from the external environment. They find that the α helix seals the center at the quasi 3-

fold axes of the T=3 particle, but there is a big opening at the icosahedral 3-fold axis, which is not seen in any non-enveloped virus. The authors do not consider the possibility that nakednaviruses may have a lifestyle in which, like the hepatitis A and E viruses, they acquire an envelope without encoding for the corresponding envelope proteins. The evolution of HBV may have been to incorporate envelope proteins to go with an envelope that was already used by its ancestors. This would explain the partially fenestrated nature of the nakednavirus particle, which is seen only in the nucleocapsids of enveloped viruses.

On a different note, it would also be easier to the reader if the Figures could be referred to in the right order. As it is, Figure 2 is cited before Figure 1, Figure 5 before Figure 3, Figures S3, S4, S5 and S6 are cited before Figure S2, etc. This renders the text a bit chaotic to follow. In addition, they provide 19 supplementary Figures. Are they all necessary? Especially that the last three are not cited anywhere in the text.

Additional comments:

Nakednaviruses : I looked at the latest classification of the International Committee on Taxonomy of Viruses (ICTV, <https://talk.ictvonline.org/taxonomy/>), which lists the Hepadnaviridae as a single family within the Blubervirales order. This family is listed as having five genera: in addition to the Orthohepadnavirus and Avihepadnavirus genera mentioned by the authors, it also lists the genera Herpetohepadnavirus, Metahepadnavirus and Parahepadnavirus, all three containing viral species that infect fish or amphibia. It would be important to relate this latest ICTV classification to the one used by the authors, as Nakednaviruses do not appear in the ICTV list.

Discussion, second paragraph: the authors seem surprised that with 24% amino acid sequence identity the capsid protein of ACNDV and HBV have the same structure. They seem to forget that back in 1980, the big surprise with the publication of the southern bean mosaic virus particle (SBMV, <https://www.nature.com/articles/286033a0>) showed that the capsid protein of this T=3 virus had exactly the same fold of that of Tomato bushy stunt virus (STNV, <https://www.nature.com/articles/276368a0>), the first high resolution structure of a virus particle ever published, despite no detectable amino acid sequence identity. And that in 1985, when the first structures of particles of viruses infecting humans were published (both picornaviruses - poliovirus <https://www.science.org/doi/10.1126/science.2994218> and rhinovirus <https://www.nature.com/articles/317145a0>), it was discovered that the three independent capsid proteins (VP1,VP2 and VP3), which had no detectable amino acid sequence similarity with each other, also shared the same fold, which was the same as those of the T=3 plant viruses that had been described previously. It is also important to recall that the herpesvirus capsid is formed by structural homologs of the capsid proteins of the tailed bacteriophages, which must have evolved separately for more than a billion years, the herpes viruses acquiring an envelope and a tegument to infect eukaryotic cells, while the bacteriophages developed a tail system to inject their genomic DNA across the bacterial cell wall. These are only two examples of multiple cases that should help put the similarity in the capsid protein between ACNDV and HBV better into evolutionary context.

Response to the reviewers :

Reviewer #1):

African cichlid nakednavirus (ACNDV) is a none-enveloped virus that is evolutionary related to the enveloped Hepadnaviridae. Pfister et al. present structural insights into the organization of virus-like particles of ACNDV using electron microscopy and NMR. They obtain maps with some 3.7-3.9 Å resolution, which allow them to model large parts of the assembly domain of the capsid protein. Their structural findings confirm the previous structural model by Lauber et al. 2016 at 8 Å resolution, in which the arrangement of the subunits in a T=3 capsid was revealed and the structural similarity to the human Hepatitis B core Antigen (HBc) was established. However, the better resolved maps allow Pfister et al. to derive structural models, which show distinct differences in the stabilization of the inner and inter dimer packing of the subunits in the capsid compared to human HBc. One interesting aspect of their work is that they observe a significant shrinkage of the capsids by lowering the pH to 5.5 which they speculate might be “reminiscent of structural changes upon viral cell entry”.

The structural work broadens our understanding of the organization of ACNDV capsids and is a valuable add-on to the existing knowledge. One strength of the manuscript is the detailed description of the methods and materials, which document the differences in the handling of the sample for NMR and for cryo-EM and at pH 5.5 and pH 7.5.

→ We thank the referee for the encouraging comment.

The introduction reads very exciting and appealing to a wide audience and phrases the fundamental question for the structural adaptation to a non-enveloped life style in nakednaviruses. However, these expectations are not met by the structural findings in the results part. The discussion and final conclusions address the questions from the introduction again but the discussion remains largely speculative and not well supported by the experimental findings.

→ We changed parts of the discussion and conclusion to accommodate the comment by the referee (see version with highlighted changes in the text).

Similarly, the conclusion about the lower charge in the interior enabling uncoating earlier in the viral life cycle is not supported by any experimental evidence. It is not even known whether an earlier uncoating takes place in ACNDV. This disconnects the solid results from the highly speculative elements of the discussion and conclusion.

→ We changed this part as shown below in the detailed discussion.

Similarly disconnected is the use of NMR and electron cryo-microscopy for the structural investigations. It is not always clear whether the NMR-studies and the electron cryo microscopy give the same findings.

→ We changed the manuscript at a number of places (including the abstract) to make it more clear which information comes from NMR and EM, respectively.

whether they are complementary or whether they even contradict each other. For example, NMR shows aggregation over time and at pH 5.5 and electron microscopy shows that there is no aggregation.

→ We apologize that the terms used were confusing although established in the literature. What was called aggregation in the NMR spectra has now been renamed to denaturation. This is what is observed in NMR and it refers to a partial loss of secondary structure and concomitant disorder in the

3D structure. This leads to newly appearing broad lines in the NMR spectrum. In EM we observe that the capsids stick together which, however, does not influence significantly the molecular structure and the NMR spectrum. This effect and only this one, is now called aggregation.

Assigned start- and end-points of the helices differ in NMR and in electron microscopy. While these discrepancies are not a problem per se, they add concerns about the statements drawn on packing of side chains, structural dynamics or pH dependent stability and conformational changes.

→ The start and end points of helices sometimes differ by one residue (Figure 1). The evaluation of the secondary structure has slightly different phi/psi angle cutoffs in different software; and also in NMR secondary-chemical-shift evaluation used to establish secondary structures. If these values are slightly off, as can be the case for first or last residues in a helix, this can lead to discrepancies as one software will recognize it as helical and the other one not. We decided not to manipulate these cutoffs to obtain a consistent picture, but preferred to keep standard values. When comparing NMR and cryo-EM (or X-ray) structures, there are many examples for this effect in the literature e.g. [10.3389/fmolb.2019.00100](https://doi.org/10.3389/fmolb.2019.00100). Thus, these discrepancies are more likely due to slightly different cutoffs and should not lead to questioning the NMR results in general.

More specific comments

Introduction

1. Line 65-72 “likely to employ fundamentally different infection and virion release mechanisms “ It was shown that naked human HBc particles are secreted in an Alix dependent manner (Bardens et al. 2011 Cell Microbiol). Therefore, it is conceivable that an envelopment independent capsid release mechanism is encoded in the capsid structure of human HBc. It is similarly likely that such a release mechanism is structurally preserved. Other capsid specific mechanisms of the viral life cycle such as the need for genome-release, the transport of capsids within the cell or the escape from the endosome are very similar for enveloped and none-enveloped Hepadna- and Nakedna viruses. Taken together, it is probably less likely that the mechanisms in nakedna viruses are so fundamentally different as suggested.

→ This is an interesting point. We toned down the previous statement and added in the discussion (p3): On the other hand, apart from enveloped virions also naked HBV capsids are released from infected cells via a different, Alix-dependent pathway [17]. The role of these naked HB nucleocapsids is not fully understood, however, it has been suggested that the naked nucleocapsids may be involved in the transmission of the viral genome [17]. Hence, naked HBV capsids might still have retained the ability to export naked capsids from nakednaviruses, which should be associated with a conservation of the capsid structures. The enveloped HB virions however are considered as the main infectious species.

Results:

2. Lines 143-145 “ Information about helix positions and lengths allowed for a fast de novo building of an initial structure”
Start positions and the end positions of the helices and the length of the helices derived from NMR and cryo-EM are somewhat different.

→ The start and end points of helices sometimes differ by one residue (Figure 1). The evaluation of the secondary structure has slightly different phi/psi angle cutoffs in different software; and also in NMR secondary-chemical-shift evaluation used to establish secondary structures. If these values are slightly off, as can be the case for first or last residues in a helix, this can lead to discrepancies as one software will recognize it as helical and the other one not. We decided not to manipulate these cutoffs to obtain a consistent picture, but preferred to keep standard values. When comparing NMR and cryo-

EM (or X-ray) structures, there are many examples for this effect in the literature e.g. [10.3389/fmolb.2019.00100](https://doi.org/10.3389/fmolb.2019.00100). Thus, these discrepancies are more likely due to slightly different cutoffs and should not lead to questioning the NMR results in general.

Out of curiosity: is this information from NMR so much more beneficial for denovo model building than the existing predictions from homology modelling and the secondary structure predictions?

→ This depends. For example, in our previous work on a-synuclein fibril structure determination, no homology models or structure predictions existed, and NMR-derived secondary structures were essential, since they allowed, amongst other, to assign the fibrils to the corresponding sample and to position an amino-acid stretch in an isolated density (Guerrero-Ferreira et al. *ELife* (2019)). Also, in Sborgi et al., PNAS 2015, the NMR-derived secondary structures allowed to build a first fibrils model. In cases as here, where the protein has homologues and first low resolution models exist, it might have been easier than in our previous work, also since EM resolution increased substantially. Still, we did not systematically investigate this, and did no comparison of the different approaches. In any case, NMR adds important information content in the sense that it is a fully independent experiment, using another sample, another approach, and other instrumentation and another operator.

3. Lines 153-155 “ Apart from signs for protein aggregation,,three additional peaks were observed in the pH 5.5 spectrum, including resonances of histidine and isoleucine residues (Figure S 3D)”. Is it clear where these three differences are located in the structure.

→ No, since site-specific assignment was difficult due to the crowding in the spectrum caused by partial denaturation

Is their evidence from the cryo-EM data, which indicates a structural change close to these residues.

→ This unfortunately cannot be addressed because peaks are not assigned and thus it is not clear which of the histidines may have changed.

Does cryo-EM see any effect of His-protonation on the structure?

→ This is not evident from the structures.

4. Line 182: “the very first three residues are not visible in the cryo-EM density map” Is the C-term (N-ter?) not resolved because of the additional Gly? If so, is there any indication that the α -helix continues to the very N-terminus?

→ Looking at the low resolution map of the ACNDV capsid from Lauber, Seitz et al. (emdb-3822) which does not feature the additional glycine in the sequence, the N-ter is resolved to a similar extent as in our map and the first three residues seem to be unresolved in emdb-3822, too.

5. Lines 189-190:” Comparison of the AB dimer, formed by the quasi-equivalent conformation of chains A and B, with the CC dimer, reveals differences in the dimer interface formed by the spike helices (Figure 3E,F compared to Figure S 8F,G)” It is not clear, which differences are meant. The figure is very busy and it is difficult to see the interactions in the figure. It is unclear, which differences in the interaction network are real and which are related to modelling errors. At a resolution of 3.7-3.9 Å side chain density is not very reliable. No data is shown that shows whether the difference in the models is supported by the density in the map. Many differences map to smaller side chains. At the given resolution their rotamer orientations cannot be assigned without ambiguity. Other differences map to negatively charged residues such as Glutamate. These residues are typically not resolved in the EM-density. Therefore, the side chain

assignment of these residues is very ambiguous. In addition, the upper spike is less well resolved and does not show any continuous density. Therefore, differences in the interaction network in the upper spike region are unlikely to be mapped with confidence. To support their claim the authors should show that the differences in the model are fully supported by the map density and that there is evidence from the NMR data, which should map a different environment for these residues.

→ We agree and thus removed panels Figure 3 D,E,F and Figure S 8 D,E,F; and G because the contact are based on unreliable sidechain positions. We changed and shortened the text (p7):For the pH 7.5 and the pH 5.5 structures, superimposed chains show that chains A and B are very similar while the N-terminus and the hand region is different for chain C compared to the other two chains (Figure 3A and Figure S 8A). Comparison of the AB dimer, formed by the quasi-equivalent conformation of chains A and B, with the CC dimer, reveals that the four helix bundles forming the spikes match well when superimposing AB with CC dimers, while the hand regions appear to be shifted (Figure 3B,C and Figure S 8B,C). This rigid body like conformational shift between the quasi-equivalent capsid subunits observed in the cryo-EM structure agrees with the solid-state NMR measurements, which show few local conformational changes, as is further reported below. In contrast to HBV Cp, where 71W and 88Y sidechains interact with each other across the spike helices (Figure S 9E), only 60H is part of a spike contact but no other large, hydrophobic sidechains (W, Y, F, or H) are involved in helix-helix contacts within the ACNDV capsid spike.

6. Lines 194-195: “ Thus, the N-terminus in the CC dimer likely stabilizes a conformation in which the core fold of the capsid protein (hand region and spike helices) is rotated relative to the dimer interface”It is unclear what is meant by the “dimer interface”? Is this the 2-fold symmetry axis in the CC-dimer (intra dimer) or the interface between dimers (inter)? Please annotate the relative rotations in the figure.

→ We removed this part, see comment #5.

7. Line 198: “The N-terminus with its $\alpha+$ helix contacts adjacent helices of the partner molecule in the dimer”

How much of the inner dimer interface is contributed by the N-terminal $\alpha+$ helix compared to the core helices in the spike?

→ Each protomer in the dimer contributes an interaction surface of around $\sim 1500\text{\AA}^2$ to the interaction with the other protomer in the dimer if the $\alpha+$ helices (residues 1-8) are not included. The $\alpha+$ helix of each protomer has an interaction surface of $\sim 350-400\text{\AA}^2$ with the other protomer, suggesting that this contribution is $\sim 20\%$. (Protein-protein interactions in the AB dimer calculated with the spider server and the EPPIC server.)

8. Lines 206-207: “ We observed that the capsid diameter is smaller at pH 5.5 (21.7 nm, Figure 2 H) than at pH 7.5 (23.0 nm, Figure 2 D) yet, the asymmetric units at pH 7.5 and at pH 5.5 appear very similar.”

Variable diameters of human HBc are frequently observed in 3D-classification regardless of an external cue. In this analysis of capsid-like particles at pH 7.5 and 5.5 a very small fraction of the cryolo selected particles was retained in the final map (according to Figure S2 10% at pH 7.5 and 36% at pH 5.5... these number do not agree with the numbers quoted in the Materials and Methods. Please check!,

→ Yes, thank you. The protocol has been corrected and is now consistent.

Therefore, the question remains whether the change in diameter is related to a change in pH or to a selection of different subsets of the particle population or some other coincidence. Another complication along these lines is that the data was collected with two very different camera set-ups (K2 vs Falcon III). These cameras have different pixel sizes and are also located at different

positions in the microscope. The pixel size depends very much on the calibration and can easily have an error in the order of 5%. As this is the difference, which is observed between the capsids at pH 5.5 and pH 7.5, it is necessary to explain how the two microscope set-ups were calibrated, whether they have been calibrated with the same standard (e.g. against map of ferritin) and what the likely error in the calibration procedure was.

→ The pixel size of our Titan Krios microscopes is calibrated with graphene grids in regular intervals. The alignment of the Krios microscopes is usually stable over many months to years. In addition, the comparison of reconstructions of stable specimens, like ribosomes, obtained over course of several years confirms the calibrations and the stability of the microscope alignment (e.g. ribosome reconstructions on Falcon cameras: EMD-11562, EMD-4300, EMD-4214, EMD-4217, EMD-3748, EMD-3531). We checked the calibration values for the time of the data collections. Due to a mistake when setting up processing for data of the sample at pH 7.5 an incorrect pixel size calibration table was used for the K2 camera (initial manufacturer calibration table) which we revised to 1.07 Å/pix (on-site calibration at the time of data collection) in the current manuscript. The pixel size of the measurement at pH 5.5 remains unchanged. The reviewer was right with her/his skepticism regarding the precision of the diameter measurement. We have performed further measurements accessing the diameter as a function of pH (with all other parameters identical) and have found that there is no significant difference between the two pH values. The text has been changed (see below).

Measuring the diameter in the maps can be quite cumbersome. Therefore, the authors chose two distinct atoms in the model of the subunits in the capsid. This gives a very accurate measure of the diameter if the model is accurate. However, in table S4 Mtriage quotes a map model CC of 0.5 at 7.4 Å for the model at pH 5.5 (please note the threshold of 0.5 is decisive threshold when comparing map and model). This suggests that either the map has no resolution of $3 \times$ Å or that the model is wrong. Please check!
and improve the map to model correlation.

→ The original manuscript contained an estimation of 7.4 Å resolution at map-model cross-correlation of 0.5. This value was generated in the early phase of the work and included in the manuscript erroneously. It is not in agreement with the clearly recognizable high-resolution features of the map. The map-model FSC in the original draft was calculated with a model that contained the unstructured residues at the tip of the spikes. In addition, masking was loose, which resulted in the inclusion of unstructured nucleic acids. We have re-refined a model without the unstructured region at the spike tips and we have calculated the model-map cross-correlation with mtriage using different mask smoothing settings. With a wide, smooth mask (e.g. using mtriage mask Mask smoothing radius = 10), the FSC intercepts $cc=0.5$ at a resolution of 5.65 Å Masking with the mtriage preset (Mask smoothing radius = 5) reduces the dip in the FSC, resulting in an intercept of $cc=0.5$ at 4.37 Å. Tighter masking reduced the early drop off in the map-model FSC further as expected when unstructured nucleic acids near the capsid shell are excluded.

To claim that there is a pH dependent change in diameter the authors should 1) calibrate the microscopes to a common standard (or collect the data with the same set-up); 2) show that the change in diameter reverses when increasing the pH again. As such large changes in diameter can be mapped at some 6-8 Å resolution, it is advised to establish the changes with small scouting experiment where one preparation of ACNDV is buffer exchanged on the concentrator to pH 7.5 then exchanged to 5.5 and then back to 7.5. For all three conditions cryo-EM data should be acquired at the same microscope.

→ We performed such a scouting experiment. We collected data of ACNDV capsid at pH 7.5, pH 5.5, and pH 7.5 after exchange from pH 5.5 on the same instrument in the same session. Indeed, we could not reproduce the diameter change and thus conclude that the change in diameter is not attributed to the pH change. We have added the experimental data to the manuscript. Following passages have thus been changed:

→ We included a new paragraph in the SI:
k. Cryo-EM experiment for particle size determination

→ We changed in the Abstract:
Furthermore, we compared ACNDV capsids at pH 7.5 to capsids at pH 5.5 whereas no significant conformational changes were observed upon decreasing the pH although the capsids are more stable to denaturation at pH 7.5.

→ We added in the Results section:
We observed that the asymmetric units at pH 7.5 and at pH 5.5 remain very similar – yet the capsid diameter is slightly smaller in our pH 5.5 model (about 22 nm) than at pH 7.5 (about 23 nm). However, an additional pH-shift experiment revealed that this change in diameter is not related to the pH change but is rather a consequence of the range of diameters present in the different capsid samples prepared for the pH 7.5 and pH 5.5 experiments (see Figures 2A, 2D, S new Figures S12, S13, S14, S15).

→ We removed in the Discussion section the part on the diameter change.
→ A figure summarizing the results of the pH shift experiments has been added to the supplementary information in addition to cryo-EM image processing schemes for the three samples:

9. Lines 208-209: “Chain B can be superimposed almost perfectly at either pH, while chains A and C are slightly shifted relative to chain B.”
Is “perfectly” because chain B was superposed with Matchmaker? Would the conclusion differ if chain C would be superposed? Please clarify (e.g. in the figure legend for 2k)!

→ If the matchmaker pair ss command on all atoms is used, the ASU overlays as shown in the image. If only chain C is superimposed, chains C would superimpose almost perfectly while chain A and B of the pH 5.5 model are shifted relative to their counterparts at pH 7.5. We clarified this in the text and in the legend.

10. Lines 230-231: "RMSD of CA atoms comparing superimposed AB and CC dimers with largest RMSD values at the hand regions. The RMSD seems to be very much dependent on how the superposition was done. If the match would be optimized for the hand region the rmsd values would probably be much smaller and in the Spike region within the error of modelling (1-2 Å). Similarly, if the AB- and CC-dimers could be fitted into a common molmap of a dimer before calculating the RMSD values, they are probably also much lower. This different way of fitting to a common map would generate a more equal distribution of errors and would probably hint at a less stunning difference of the hand-region.

→ We agree that the exact RMSD values depend on how the superposition is done. We compared different methods to superimpose CC and AB dimers: 1) align command in ChimeraX with all molecules (excluding hydrogens), 2) align command with only the hand regions (residues 101 onwards). 3) Fitting of the CC dimer into the molmap of the AB dimer. In cases 1 and 3 RMSD values are highest at the hand region and in case 2 RMSD values are highest at the loop between helix 2 and helix 3 followed by the hand region. Overall, different matching does not change the conclusion that the main difference in AB vs. CC dimers is a slightly different orientation of the hand region relative to the rest of the molecule. In our opinion, this is already visible in the superposition in panel B and we thus removed panel C since the RMSD plotting implied some sort of numeric precision that is not appropriate (because the precise numbers depend on the matching algorithm as the referee pointed out)

11. Lines 272-273: "Unassigned cryo-EM map density located below the spikes of AB dimers but not below CC dimers was found in the ACNDV capsid maps (Figure S 11). Likely sources of the diffuse density below AB dimers are enclosed RNA (solid-state NMR identifies nucleic acids to be RNA and not DNA [data not shown]) and/or the flexible CTD of the capsid protein" The cryo-EM density is shown in Figure S12!

→ We thank the referee for catching this and apologize for wrong figure numbering.

and appears very speckled. Therefore, it is unclear what the relevance of this density pattern is. This should probably be low-pass filtered for better visibility.

→ We have generated a new figure S 17, using the same maps as in the original figure, but filtered using the local resolution filter of cryoSPARC. Local resolution filtering preserves the structural high-resolution details of the capsid shell and in addition - as expected - shows continuous density (at lower resolution) at the spike tips and inside the capsid. The new figure shows a slice through the capsid exterior perpendicular to the surface (at pH 7.5 and pH 5.5) with clearly visible density below the AB, but not CC dimers. In addition we show in more detail a slice through the capsid map (at pH 7.5) with fitted AB and CC dimers.

Considering that charges, RNA and DNA become a major part of the discussion, the following points should be clarified:

1) How much RNA is incorporated in the capsid shells per protein. It should be possible to estimate the ratio from Agarose gels stained for RNA and protein. Alternatively, the ratio can also be compared spectroscopically. Is the ratio different to human HBc?

→ We calculated the RNA/protein ratio by UV/vis spectroscopy based on the procedure described in the paper of Porterfield and Zlotnick <https://doi.org/10.1016/j.virol.2010.08.015>, and found about 2600-2800 nucleotides per capsid. But we cannot rule out that the RNA content decreased over the two years of sample storage (between EM and UV/Vis) and we therefore decided not to use this measurement in the manuscript.

2) Does the EM-map give a radial density profile, which shows that incorporated density aggregates in distinct shells?. If so, what is the distance between the peak of the protein shell and the “RNA-shell”. For human HBc the distance is very characteristic and there is a clear gap between protein shell and “RNA-shell”. This reviewer expects that in ACNDV the distance is smaller because many of the positive charges are accumulated on the inner capsid surface (see Figure S13) underneath the spikes rather than in the unresolved CTD. There could also be other mechanisms of packaging and charge compensation which might generate a more diffuse density distribution in the capsid interior.

→ Indeed, we also see a gap between protein shell and RNA shell. We find, at pH 7.5, maxima at 11.6 nm and 9.2 as compared to the T=4 values of HBV of 13.5 nm and 10.5 nm. Seitz S, Urban S, Antoni C, Böttcher B. 2007. Cryo-electron microscopy of hepatitis B virions reveals variability in envelope capsid interactions. *EMBO J* 26:4160–4167. doi:[10.1038/sj.emboj.7601841](https://doi.org/10.1038/sj.emboj.7601841). The difference between the two maxima is 2.4 nm, indeed smaller than the 3 nm found for HBV. We have now added the radial density plots to Figure S 14. It should also be noted that there is a slight difference for the two pH values.

12. Lines 287-288:” a quicker aggregation within the NMR rotor suggests that pH 5.5 capsids are less stable than pH 7.5 capsids.” This statement confuses solubility with stability. The authors observe solubility. They show in their supplemental material, that negatively stained capsids in “aggregated” samples look intact without indication for denaturation. They are probably even able to dissolve the aggregates again as they do not see aggregation at the lower concentrations required for negative stain. The authors do not show a reduced stability of the capsids. To show a change in stability they should do a Thermo-shift-assay, which shows a different Temperature for the onset of denaturation. To relate stability to a folding enthalpy, more sophisticated experiments such as differential scanning calorimetry would be necessary. This reviewer does not understand how an observation of solubility relates to possible conformational changes or decapsidation in the subsequent discussion.

→ There is a confusion with the term aggregation which is differently used in different fields. What was called aggregation in the NMR spectra of the original manuscript has now been renamed to denaturation (and is related to stability). This is what is observed in NMR and it refers to a partial loss of secondary structure and concomitant structural disorder in the 3D structure leading to broad lines. In EM we observe that the capsids stick together which, however does not influence significantly the molecular structure and the NMR spectrum. This effect (and only this one) is now called aggregation throughout the manuscript.

→ We agree that more sophisticated experiments would give more insight, this was however out of the scope of this study. Here, we wanted to state that particles lose the characteristic alpha-helical fold faster at lower pH when kept in the NMR rotor which is linked to the stability of the capsid. This denaturation behavior is rather untypical: many proteins are very stable in the sediments [Wiegand et al, <https://doi.org/10.3389>]. We see in the NMR spectra of pH 5.5 samples (Figure S3) that a portion of the proteins loses its characteristic alpha-helical structure since peaks are shifted towards shifts that are typical for beta-sheets. The text has been adapted.

13. Lines 292-300: “ The evolutionary lineages...”In the context of evolutionary lineages the authors should interrogate the structure for the inter-dimer contacts, which have three conserved motives according to Dill et al. J. Vir 2016. Whereas the other land mark features such as inner dimer contact, tips of the spike, C-term, N-term are not conserved between Hepadnaviridae including Nakednaviridae,

→ We made this clearer and added the following: Three core motifs: motif I (between helix α 1 and α 2), motif II (at the bottom of helix α 3), and motif III (hand region) are known to be conserved in the amino acid sequences of *Hepadnaviridae* [Dill et al]. Comparison of the motifs in the capsid structures of HBV, DHBV, and ACNDV (Figure S 9) shows that these three regions are structurally conserved among the representative members of hepadnaviruses (HBV and DHBV) and nakednaviruses (ACNDV), which is in line with the sequence alignment by [Lauber, Seitz et al]. The inter-dimer contacts (motif I and III) thus seem to be particularly conserved, while the C-termini, N-termini, spike tips, and intra-dimer contacts such as cysteine bonds are not conserved among hepadnaviruses including nakednaviruses.

14. Lines 316-319: “Plausibly, the reduced number of arginine sidechains may be unable to electrostatically compensate for a largely double-stranded full-length DNA such that the capsid shells are disrupted early during second strand DNA synthesis; conversely, particles containing shorter reverse transcription products, such as from the major 2.0 kb SP1 splice product, would still be stable.” This discussion of the second strand formation is purely speculative. The mechanism of charge balance in ACNDV is unknown. Charges could also be compensated by incorporating divalent cations such as Magnesium or small molecules like spermidine. Human HBC inhibits the interaction of the C-term with unspecific RNA by phosphorylation. However, there are much less serins in the ARD of ACNDV (3) compared to human HBC (7), which could be potentially phosphorylated. Therefore, the charge balance mechanism might vary rather than modulating the second DNA strand formation or disrupting the capsid. It is also unclear what the importance of disrupting a capsid during formation of the 2nd DNA strand would be. This would basically interfere with the newly formed virus before it egresses from the infected cell. However, decapsidation would make more sense for a mature capsid after it entered the cell and has targeted the nucleus. Without any experimental evidence for a formation of a shorter second DNA-strand in ACNDV, this discussion is too speculative to be useful – rephrase and support by evidence (either experimental or via references)

→ We simplified the discussion in order to avoid speculation (p13): To date, the charge compensation mechanism of nucleic acids in nakednaviruses is largely unknown. Incorporation of additional charges such as cations may help to add to the charge compensation of ACNDV capsids *in vivo*. While we have only investigated *in vitro* Cp we still assume that they are the dominant species also *in vivo*.

15. Lines 329-333:” Still, the packaging capacity would be smaller for ACNDV Cp because it exclusively assembles into T=3 capsids which means that 25 % less protein is available for neutralization of nucleic acid phosphates. Since genomes of HBV and nakednaviruses are of comparable size of around 3’000 bp, the genomes of nakednaviruses appear too long to form stable nucleocapsids and thus, reverse transcription might cease at an earlier stage in ACNDV nucleocapsids as compared to HBV.” This is an interesting aspect and very much along lines 316-319. However, it is not known whether ACNDV also forms exclusively T=3 capsids in mature viruses. It could well be that the viral polymerase together with the RNA-genome nucleate the assembly of larger capsids such as the T=4 capsids. Again, 2nd strand synthesis could be tested in cellular assays.

→ It is indeed not known whether T=4 nakednaviruses exist *in vivo* but this seems rather unlikely since the T=3/T=4 ratio in HBV is very similar when comparing recombinantly expressed particles with isolates. [Dryden 2006 <https://doi.org/10.1016/j.molcel.2006.04.025>, Wu, Watts, 2020 <https://doi.org/10.1371/journal.pcbi.1007782>].

16. Lines 340-341: “Also, the smaller opening of the ACNDV capsid triangular pores (in the center of Figure 6A) are compensated by bigger openings at the threefold axes”. Whether something can pass holes very much depends on the properties of the holes in addition to the size. What are the properties of the holes at the three-fold axes (e.g. hydrophobic, hydrophilic, charged?), Are these properties any different in human HBC? Figure 6A is one of the few places in the manuscript, where the map density is shown. The threshold is chosen that the map looks very skinny and the holes might appear exaggerated. The holes look somewhat larger than in the space fill representation of the model in Figure 2. It would make sense to check the properties of the holes at the local and strict 3-fold symmetry axes for example with the programme “HOLE” to get further insights into the significance of the difference in apparent hole size.

→ We added following passage to strengthen our point (p13): Analysis with the HOLE software [49] yields a hole radius of 1.1 Å for the local 3-fold axis of the ACNDV capsid at the narrowest point, while the equivalent holes measure 4.8 Å for the HBV and 6.1 Å for DHBV capsid. The 3-fold axis, on the other hand, takes a minimum radius of 6.1 Å in the ACNDV capsid and 4.8 Å in HBV; the two-fold axis in DHBV has a minimum radius of 3.3 Å. [...]In conclusion, diffusion of small molecules through the ACNDV capsid will hardly be limited by the occlusion of triangular pores by the $\alpha+$ helix.

Lines 474 -478: “Cp-like large and accessible hydrophobic area within the capsid spikes in ACNDV Cp implies that the hydrophobic pocket in HBV Cp evolved after the separation of hepadna- and nakednaviruses. While still hypothetical, binding of a natural pocket factor into the HBV Cp hydrophobic pocket may be related to the capsid maturation and envelopment [48, 55,56]. Hence, the absence of an equivalent pocket in ACNDV Cp might well reflect that the HBV Cp hydrophobic pocket co-evolved with the acquisition of envelopment. The authors compare structural elements of ACNDV mainly with human HBC. They find that many important structural determinants are not conserved, such as the stabilization of the dimer interface, the disulphide bind and an enigmatic hydrophobic pocket in. However, the side-by-side display of the monomers of DHBC, HBC and ACNDV in Figure 5 suggests that the packing of the helices in the core spike without the extension is much more similar to DHBC than to HBC. In particular helix 3 is tilted away from the dimer axis in DHBC and in ACDNV but not in HBC and helix 4 is straight in ACDNV and DHBC but kinked in HBC. Like ACDNV, DHBC-dimers are not stabilized by an SS-bridge, their inner dimer interface is more hydrophilic, they have a $\alpha+$ helix, although in a different orientation and they do not have a hydrophobic pocket. It would be helpful to extend the comparison in a similar way towards DHBC (e.g. add a comparable panel for DHBC to figure S14), In the light of a missing hydrophobic pocket in DHBC, this feature seems to be unrelated to envelopment in general. The analysis of Lauber et al. does not suggest that ACDNV is more similar to HBC than to DHBC.

→ We changed the text to match with the conclusion here. Providing the DHBV images in the SI is a good idea. We changed the text as follows (p18): However, the hydrophobic surface shows that ACNDV Cp lacks a large hydrophobic interface between adjacent monomers as in HBV Cp (Figure S 14), hence the unassigned density more likely corresponds to bound RNA or the flexible CTDs. While still hypothetical, binding of a natural pocket factor into the HBV Cp hydrophobic pocket may be related to the capsid maturation and envelopment [56, 63, 64]. However, compared to HBV, DHBV Cp does not feature an equally large hydrophobic area at the dimer interface but displays two small hydrophobic patches shown in Figure S16: one at the extension domain and one towards the spike base.

Conclusions:

17. Lines 484-485 Conclusion: “ although envelopment implies fundamental differences in virus lifestyle, including host and tissue tropism, its acquisition had only a minor effect on the capsid protein fold. Most of the differences in host and cell tropism are related to the envelope and not to the

structure of the capsid. Egress of non-enveloped capsids is maintained in HBC (see above). Therefore, the differences in virus-life style in respect to the capsid are probably not so fundamental as suggested by the authors. Thus, it is unsurprising that the general fold of the capsid is maintained.

→ We agree with the statement that the host and tissue tropism is related to the properties of the newly acquired envelope. Still we find it unexpected that the capsid structure is so highly conserved despite the fact that some functions are no longer necessary in the enveloped virus while others are added. See also answers to reviewer 2.

18. Lines 487-490: "Given the similar genome sizes of about 3 kb of both nakedna- and hepadnaviruses this implies that formation of the double-stranded DNA genome inside the capsids exerts more internal pressure on ACNDV than HBV capsids, hence genome uncoating may take place earlier in the viral life-cycle of nakednaviruses." I do not understand how this can be concluded from the structural work considering that 1) It is unknown whether T=3 capsids are the predominant form for mature virus particles; 2) whether DNA strand formation leads to decapsidation. What is meant by earlier in the viral life-cycle? Does this mean that decapsidation takes place in the cytosol before the release of the mature virus from the cell? This does not seem to be a plausible model that is in any way supported from the structural findings.

→ We have changed this sentence as follows: Given the similar genome sizes of about 3 kb of both nakedna- and hepadnaviruses this implies that additional positive charges may be taken up into the capsid to compensate for the negatively charged nucleic acids.

19. Lines 490-491: "the assembled icosahedral viral particles were found to be, in vitro, less resistant against denaturation than HBV, as seen by NMR." The authors show that the capsids are less soluble but not less resistant against denaturation. In addition, the negatively stained controls at a much lower concentration suggest that the aggregation is reversible by dilution, as they do not observe it. This again points towards solubility rather than denaturation.

→ There is a misunderstanding regarding the interpretation of the NMR experiments. These samples are in a sediment and the solubility plays no role. In fact, the sediment behaves fully like a crystalline solid in NMR experiments. Sticking together of capsids is not significantly changing the spectra. Denaturation refers, in this context, to significant changes to the secondary structure (and of course the 3D structure).

→ In order to further investigate the stability of the ACNDV capsids reaction to a treatment with guanidine hydrochloride (GuHCl) we performed negative stain TEM imaging. TEM images of the samples (Figure S 6) show that capsids in buffer containing 0.3 M GuHCl remain intact. However, with 0.5 M GuHCl capsid preparations become inhomogeneous with particles adopting a broader variety of different shapes. With 1.0 M no intact capsids can be identified anymore. For comparison, HBV Cp seems to be able to withstand higher concentrations of GuHCl than ACNDV Cp. It was for example found that GuHCl concentrations smaller than 1.25 M were insufficient for HBV capsid disassembly [41]. This finding supports the NMR data.

→ We added in the Conclusion (p19): In general, the assembled icosahedral viral particles were found to be less resistant than HBV capsids against denaturation with respect to time in storage, low pH as detected by NMR, and as well as with exposure to GuHCl as seen by TEM.

Materials and Methods:

20. Lines 573-576 Repetition of the start of the previous paragraph (lines 555-558).

→ This section has been updated.

Validation

Reports

The validation reports are incomplete (no clash report). The reports do not show the map to model validation. The validation reports shows that residues 56-75 are missing, whereas only 66-75 were reported missing in the results of the paper > The validation does not match the represented results.

→ We recalculated the model, all data has been redeposited to the respective databases.

Supplemental Material

21. Table S4 The table indicates a low similarity between map and model for the pH5.5 structure (CC 0.5 map vs model = 7.4 Å). This is unexpectedly bad. Show a detailed presentation, which supports the quality of the model and its agreement with the map.

→ Yes, this was a mistake. We updated the table.

It would be nice to either extend the table or have another table for the details on the imaging and image analysis (microscope, detector, magnification pixel size, focus range, total exposure, exposure rate, linear vs counted, number of selected particle, number of particles in final map, resolution according to FSC...)

→ We apologize for the somewhat confusing presentation of the cryo-EM information. We have added a table that contains the key cryo-EM data collection and image processing data as suggested by the reviewer. We also have added FSC curve plots of capsid datasets at pH 7.5 and pH5.5 processed with icosahedral symmetry and we have added processing information for the three datasets of the newly added pH shift experiment.

Out of curiosity: Did the Krios with K2 had no energy filter? If not add information on the energy filter setting

→ We apologize for the omission. Our Titan Krios is equipped with an energy filter and we used it with a slit width of 20eV. We have added this information to the data acquisition parameters table.

22. Figure S2 The figure is too small to recognize the details. > Make two separate figures for pH 5.5 and 7.5.

→ We have generated new SI 6-8 figures for the image processing flowthrough for the pH 7.5 and pH 5.5 capsid and we have redrawn the figure showing the local resolution of the asymmetric unit. The new figures contain resolution maps and FSC curves for all maps.

The numbers of particles do not match the numbers in the materials and methods in the main paper... check and correct.

→ Yes, we corrected it. The numbers in the text were from a different (older) reconstruction.

FSC-curves are missing for documenting the quality of the maps.

→ We have included FSC curves of all reconstructions shown now, including ACNDV capsid at pH 7.5 with icosahedral symmetry and after symmetry expansion with focused classification of the asymmetric unit, ACNDV capsid at pH 5.5, as well as the three datasets of the pH shift experiment that we have newly added to the manuscript. We have included a panel with resolution maps for all maps.

Gallery of details of map with fitted model are missing for documenting the quality of the map and the fidelity of the model.

→ We have added a figure S 9 that shows the map quality.

23. A8 D-F This is very busy and hard to see. Try thinner slabs.

→ We decided to not show it since sidechain positions are associated with some uncertainties as the referee pointed out

24. S10 Try local resolution filtering of the map (or deep learning density modifications) for a more adequate representation of the upper spike region.

→ We have prepared a new figure that uses maps that have been local resolution filtered. The connection of the spike helices are clearly visible in those maps. Attempts to further improve the density in the spike region by focused 3D classification without alignment were not successful. We have generated a new figure that shows the upper spike region as well as the extra density under the spikes (S17, please see above).

25. Figure S12 This figure shows that many side chains in the model are not accommodated by the EM-density. This raises concerns about the validity of the side chain orientations in the model and subsequently the interaction/contact networks shown in Figure 3 and S8. At a resolution of only somewhat better than 4Å, a discussion of changes in such interaction networks seems to be overstating the structural information.

→ We agree and changed those figures.

Reviewer #2 (Remarks to the Author):

The manuscript by Pfister et al builds on previous work by co-authors Stefan Seitz and Ralf Bartenschlager on the evolution of the hepatitis B virus (HBV) and its relation to naked viruses with a very similar capsid and the same genome replication strategy. The previous work had reported a cryoEM structure at 8Å resolution of the African cichlid nakednavirus (ACNDV) using recombinant capsid protein produced in *E. Coli*, which spontaneously assembles into icosahedral capsids with triangulation number $T=3$. Here, the authors used a combination of single-particle cryo-EM to a resolution of 3.7Å and magic-angle-spinning solid-state nuclear magnetic resonance (MAS ssNMR) to provide a full atomic model of the ACNDV capsid. The authors attempt to identify potential clues to understand how the naked capsid of ACNDV might deliver the viral genome into the target cells nucleocapsid, a process that in the case of HBV takes place via membrane fusion, which in principle is not possible for a naked virus particle. This leads them to determine the structure of the recombinant ACNDV capsid at pH 5.5 as well as pH 7.5 (accompanied by determination of the MAS-ssNMR structure), which shows that the particles shrink at acid pH to a diameter of 217Å instead of 230Å (close to 6% shrinkage, which means an 11% reduction in viral surface area, and 16% shrinking of the total enclosed volume. Yet they do not observe any particular role of pH in particle destabilization, which could be a way of disassembly upon uptake into an endosome. They observed however a tendency of aggregation of the particles at low pH, which they can also see in the MAS-ssNMR experiments. The authors also examine the particles at pH 9 by MAS-ssNMR, without observing any major differences.

Overall, the results reported are important as they provide an atomic model for the Nakednaviruses that was not available. I have, however, issues with the way the manuscript is presented, as the authors do not seem to use the resulting model correctly in order to try to identify all potential clues for virus entry they claim to be searching.

→ We address these points in the more detailed comments below.

Also, the text appears at moments too technical, as if the paper was destined to specialists in the techniques being used (in particular, NMR), rather than to the average reader of Nature Communications.

→ We introduced some more didactical information as addressed below.

In particular, the fact that NMR is capable of providing information on the dynamics of a particle, which are absent in static structures such as those obtained by X-ray crystallography or by cryo-EM, is not presented at all.

→ We now shortly address that in the manuscript

For instance, the reader is shown ^{13}C - ^{13}C DARR correlation spectra or hNH spectra, without any attempt to introduce them to the reader. The paper would greatly benefit from explaining the rationale for the dual approach taken for the structural studies, that the resolution of 3.7Å is limiting in the side chain assignment to build a full model, and that the complementarity introduced by NMR is very important to obtain a reliable atomic model in this case.

→ We have taken this comment into account and have explanatory text in several places.

Also, that standard solution NMR is not applicable to study particles of the size of the ACNDV capsids, but that the powerful methods of solid-state NMR measure while spinning the sample at high speed at the magic angle makes it feasible. They should also explain that like in solution NMR, it is necessary to use certain isotopes (in particular ^{13}C), and that DARR stands for Dipolar Assisted Rotational Resonance, in which the magnetization is transferred from hydrogen (^1H) to ^{13}C nuclei, from where it is further transferred to other ^{13}C nuclei which are close in space, in a process that allows to build an atomic model from the resulting spectra. And that hNH spectra inform on the magnetization transferred from ^1H to ^{15}N via cross-polarization, which is also important to build the model. It is not necessary to give a full course on NMR, but just enough explanation so that the reader understands the meaning of the spectra that are shown.

→ We added the following to the results section in order to better explain the NMR experiments (p5): Liquid-state NMR is not suitable for such large particles because the molecular tumbling is too slow, and we employed magic-angle spinning (MAS) solid-state NMR for an initial characterization of ACNDV Cp. All amino acids were uniformly ^{13}C - ^{15}N labelled. The capsids were sedimented into MAS rotors by ultracentrifugation alleviating the demand for crystals [32, 33]. The 2D ^{13}C - ^{13}C correlation experiment with a DARR (dipolar-assisted rotational resonance) [34] mixing time of 20 ms for magnetization transfer in ACNDV capsids at pH 7.5 displays a 2D-fingerprint spectrum typical for an α -helical protein (Figure S 1C). [...] Furthermore, amide and HA hydrogens were assigned and provide a starting point for future experiments e.g. for protein dynamics characterization [38]. The proton-detected hNH spectrum, which displays the correlation of hydrogen and nitrogen resonances, is shown in Figure S 3A.

In addition, they should highlight the impact of information given by split peaks for instance, where there are different interactions in the various environment of a given residue within the capsid. As it is, the authors mention that the tryptophan residues of the ACNDV capsid give rise to split peaks but they do not explain what information it is providing about the dynamics of the structure, or about alternative conformation on the three different environments of the T-3 capsid,

→ We added more information when peak splitting is mentioned for the first time (p10): Peak splitting may occur for quasi-equivalent protein chains with small structural differences due to local adaptation of the chains to the icosahedral symmetry as was previously observed with HBV Cp [22]

In addition to the issues related to the presentation of their results, the authors do not seem to exploit the atomic model to get extensive information. For instance, does any hydrophobic patch appear on the capsid in the pH 5.5 form that is absent in the pH 7.5 form, which could explain the observed aggregation?

→ No, the structures are nearly identical at both pH values and we do not see a significant difference in hydrophobicity.

They do show the surface hydrophobicity distribution for an AB dimer in Fig. S14, but it is on the whole particle surface that this should be shown, if they are to identify a potentially functional patch.

→ We updated this part (p13) and added a new SI figure (S 20): The hydrophobic/hydrophilic outer surfaces of ACNDV, HBV, and DHBV Cp show a similar pattern (Figure S 20). A difference can be seen around residues 25L and 87L (ACNDV numbering) where ACNDV Cp seems to be slightly more hydrophobic than HBV Cp; in DHBV Cp this region is partly occluded by the extension domain. The outer surface of ACNDV is more positively charged than HBV and DHBV capsids (Figure S 18,S 20)

similar as at the capsid interior (Figure S 18). In conclusion, diffusion of small molecules through the ACNDV capsid will hardly be limited by the occlusion of triangular pores by the α^+ helix. This finding is quite unexpected for an unenveloped virus which could support that ACNDV may adapt a quasi-envelope without actually encoding for the corresponding surface protein, similar to what has been described in hepatitis A viruses [13, 14] and hepatitis E viruses [15]. Such a quasi-envelope would protect the genetic material from outside influences.

Similarly, can they calculate the surface electrostatic potential of the particles at the two pH values, to see if there is a different distribution in the two?

→ We have calculated surface electrostatic potential maps of the ACNDV particle at pH 7.5 and pH 5.5 and generated a new figure that shows these maps as well as a electrostatic potential map of HBV capsid.

Again, they show the surface electrostatic potential for the inner surface of the AB dimer in Fig. S13, but it is the external surface of the whole particle that would be relevant to look at.

→ A new figure was added to the SI (S 15), which shows a larger part of the capsid surface and puts the structural conservation of charged residues of both capsids into perspective.

Reviewers' Comments:

Reviewer #1:

Remarks to the Author:

The authors have appropriately addressed the comments raised in the initial review, and the paper represents an interesting structural investigation using EM and NMR. The revised version has become stronger and allows the reader to better understand technical details.

I see one point that needs to be addressed: in the abstract the authors state that "Residues at the spike tip were observed by solid-state NMR despite residual dynamics," but the assignments that are prominently promised in the abstract turn out to be "tentative": the main text states "It was possible to tentatively assign additional NMR peaks to the ACNDV Cp spike loop region (Figure 4)". At the least, the abstract shall contain the "tentative". Even better, it would be helpful to state how the peaks have been (tentatively) assigned. Is the assignment exclusively based on a remaining peak in the characteristic frequency range for a certain amino-acid type? How close are those peaks to the noise level?

Along these lines, the cross denoting the position of K69 in Figure 4 is not at the position of the peak. Why is this?

I would recommend that the authors address a few additional points (minor):

The authors used here the NMR-derived positions of helices for generating the structural models. As this approach, or variations have been used before, it seems appropriate to cite e.g. 10.1073/pnas.1507579112 , 10.1038/s41467-019-10490-9 , 10.1002/anie.201505065

The following comments refer to new text (as highlighted in yellow in the manuscript):

On page 5, the authors refer to the DARR spectrum of Figure S1C, stating that this is a typical spectrum for a helical protein. I would claim that even for an expert it is essentially impossible to confirm this statement by inspection of Figure S1C. Without annotating the spectrum (e.g. by indicating typical alpha-helical/beta-strand shifts) I would claim that Figure S1C does not help the reader assess the claim that it is a spectrum typical for an alpha-helical protein.

On page 5 the authors refer to proton-detected dynamics experiments by MAS NMR, and refer to a single paper (from their own group), which is not really a publication about dynamics. There are many publications about protein dynamics characterization, including several extensive reviews. The citation looks like a mistake here, or at least an oversight of the rich literature on the topic.

Reviewer #2:

Remarks to the Author:

Pfister et al have conveniently revised their manuscript, following the comments from both reviewers. One major difference in the new version is that they do not report any more a considerable shrinkage of the capsid at acid pH, which they had speculated could have an effect in virus entry. Re-reading the manuscript, I find that despite reporting an atomic model for the Nakednaviruses, which was lacking, the paper remains only confirmatory, and provides very little novel insight for the biology of Nakednaviruses compared to hepadnaviruses. The authors try to compensate by giving a lot of details of interactions they see, but I could not perceive what it all means for the biology of these viruses, other than multiple speculative assertions. The combination of cryo-EM and solid-state NMR is indeed a methodological plus, yet all the details described in the manuscript fall short of providing novelty.

Reviewer #1 (Remarks to the Author):

The authors have appropriately addressed the comments raised in the initial review, and the paper represents an interesting structural investigation using EM and NMR. The revised version has become stronger and allows the reader to better understand technical details.

I see one point that needs to be addressed: in the abstract the authors state that "Residues at the spike tip were observed by solid-state NMR despite residual dynamics," but the assignments that are prominently promised in the abstract turn out to be "tentative": the main text states "It was possible to tentatively assign additional NMR peaks to the ACNDV Cp spike loop region (Figure 4)". At the least, the abstract shall contain the "tentative".

Done

Even better, it would be helpful to state how the peaks have been (tentatively) assigned. Is the assignment exclusively based on a remaining peak in the characteristic frequency range for a certain amino-acid type?

Yes, in fact on a combination of two frequencies in the 2D crosspeaks.

How close are those peaks to the noise level?

They are all relatively weak with K69 being the weakest peak very close to noise.

Along these lines, the cross denoting the position of K69 in Figure 4 is not at the position of the peak.

In fact K69 is a shoulder on a stronger peak just below. The cross is at the position of K69.

Why is this?  It's a very weak peak (next to a strong peak) and it is difficult to see because it is so weak. We do mention this in the caption by saying

The peak for 69K has only low intensity and is close to a stronger peak

I would recommend that the authors address a few additional points (minor):

The authors used here the NMR-derived positions of helices for generating the structural models. As this approach, or variations have been used before, it seems appropriate to cite e.g. [10.1073/pnas.1507579112](https://doi.org/10.1073/pnas.1507579112) , [10.1038/s41467-019-10490-9](https://doi.org/10.1038/s41467-019-10490-9) , [10.1002/anie.201505065](https://doi.org/10.1002/anie.201505065)

We have added these references

The following comments refer to new text (as highlighted in yellow in the manuscript):

On page 5, the authors refer to the DARR spectrum of Figure S1C, stating that this is a typical spectrum for a helical protein. I would claim that even for an expert it is essentially impossible to confirm this statement by inspection of Figure S1C. Without annotating the spectrum (e.g. by indicating typical alpha-helical/beta-strand shifts) I would claim that Figure S1C does not help the reader assess the claim that it is a spectrum typical for an alpha-helical protein.

We have added the following to the text: key signals for the distinction between α -helical and β -sheet structures are the Alanine CA-CB crosspeak. All signal intensity is found around the α -helical shift values of 54.86/18.27 ppm and no intensity at 50.86/21.72 ppm (β -sheet). These two regions are known to be a clear indicator for secondary structure for alanine (Wang Y, Jardetzky O. 2002. Probability-based protein secondary structure identification using combined NMR chemical-shift data. *Protein science : a publication of the Protein Society* **11**:852–861.)

On page 5 the authors refer to proton-detected dynamics experiments by MAS NMR, and refer to a single paper (from their own group), which is not really a publication about dynamics. There are many publications about protein dynamics characterization, including several extensive reviews. The citation looks like a mistake here, or at least an oversight of the rich literature on the topic.

We replaced the citation by a recent comprehensive dynamics review.

Reviewer #2 (Remarks to the Author):

Pfister et al have conveniently revised their manuscript, following the comments from both reviewers. One major difference in the new version is that they do not report any more a considerable shrinkage of the capsid at acid pH, which they had speculated could have an effect in virus entry. Re-reading the manuscript, I find that despite reporting an atomic model for the Nakednaviruses, which was lacking, the paper remains only confirmatory, and provides very little novel insight for the biology of Nakednaviruses compared to hepadnaviruses. The authors try to compensate by giving a lot of details of interactions they see, but I could not perceive what it all means for the biology of these viruses, other than multiple speculative assertions. The combination of cryo-EM and solid-state NMR is indeed a methodological plus, yet all the details described in the manuscript fall short of providing novelty.

We respectfully disagree with the referee.